# OPTIMAL STRATEGIES
# AGAINST GENERATIVE ATTACKS

**Roy Mor**
Tel Aviv University
Tel Aviv, Israel

**Erez Peterfreund**
The Hebrew University of Jerusalem
Jerusalem, Israel

**Matan Gavish**
The Hebrew University of Jerusalem
Jerusalem, Israel

**Amir Globerson**
Tel Aviv University
Tel Aviv, Israel

## ABSTRACT

Generative neural models have improved dramatically recently. With this progress comes the risk that such models will be used to attack systems that rely on sensor data for authentication and anomaly detection. Many such learning systems are installed worldwide, protecting critical infrastructure or private data against malfunction and cyber attacks. We formulate the scenario of such an authentication system facing generative impersonation attacks, characterize it from a theoretical perspective and explore its practical implications. In particular, we ask fundamental theoretical questions in learning, statistics and information theory: How hard is it to detect a "fake reality"? How much data does the attacker need to collect before it can reliably generate nominally-looking artificial data? Are there optimal strategies for the attacker or the authenticator? We cast the problem as a maximin game, characterize the optimal strategy for both attacker and authenticator in the general case, and provide the optimal strategies in closed form for the case of Gaussian source distributions. Our analysis reveals the structure of the optimal attack and the relative importance of data collection for both authenticator and attacker. Based on these insights we design practical learning approaches and show that they result in models that are more robust to various attacks on real-world data.

## 1 INTRODUCTION

Generative models have attracted considerable attention since the introduction of Generative Adversarial Networks (Goodfellow et al., 2014a). Empirically, GANs have been shown to generate novel data instances that resemble those in the true distribution of the data. The success of GANs also comes with the risk that generative models will be used for attacking sensor-based security systems. One example is identity authentication systems, where an individual is identified via her images, and GANs might be able to generate such images to gain access (Thies et al., 2016). Another is anomaly detection systems protecting critical infrastructure. As demonstrated by recent cyber-attacks (notably the Stuxnet attack) sensors of these systems can be hijacked, so that GANs can be used to generate "normal" looking activity while the actual system is being tampered with. The latter is, in fact, a new form of a man-in-the-middle attack.

Our goal here is to construct a theoretical framework for studying the security risk arising from generative models and explore its practical implications. We begin with a simple key insight. If the attacker (i.e., the generative model) has unlimited observations of the source it is trying to imitate, it will be able to fool any authenticator. On the other hand, if the attacker has access to fewer sensor observations than the number of fake observations it needs to generate, it seems intuitively clear that it cannot always succeed (as we indeed prove in Sec. 4). Therefore, the optimal defense and attack strategies depend crucially on the amount of information available to the attacker and authenticator.

Motivated by the above insight, we cast the authentication setting as a two-player maximin game (authenticator vs. attacker) where all observations are finite. Specifically, there are three key obser-

vation sets to consider: those available to the attacker, those that the attacker needs to generate, and those available to the authenticator when designing the system. Our goal is to understand how these three information sources determine the optimal strategies for both players. Under the realistic assumption that cyber attackers are sophisticated enough to play optimal or close to optimal strategies, a characterization of the maximin authentication strategy can be of significant value.

We prove several theoretical results characterizing the optimal strategy for both players. These results highlight the role played by the available observations as well as the functional form for an optimal attacker and authenticator. We refer to the setting above as "GAN in The Middle" (GIM) due to its similarity to "man in the middle" attacks. After describing our theoretical results, we show how to learn both authenticator and attacker policies in practice, where both are based on neural architectures. Our GIM method can be applied to multiple practical problems. The first is building an authentication system that is robust to impersonation attacks. The second is building a data generating mechanism that can generate novel data instances. Finally, we evaluate the method empirically, showing that it outperforms existing methods in terms of resilience to generative attacks, and that it can be used effectively for data-augmentation in the few-shot learning setting.

## 2   PROBLEM STATEMENT

We begin by motivating our problem and formulating it as a two-player zero-sum game. As a simple illustrative example, consider a face authentication security system whose goal is to maximize authentication accuracy. The system is initialized by registering $k$ images of an individual $\theta$ (the "source"), whose identity is to be authenticated. At test-time, each entity claiming to be $\theta$ is required to present to the system $n$ of its images, and the authentication system decides whether the entity is $\theta$ or an impersonator. We let $m$ denote the maximum number of "leaked" images any attacker obtained. We observe that if an attacker obtained $m \geqslant n$ images of $\theta$, it can present $n$ of those images. Thus the observations generated by the attacker are indistinguishable from ones generated by $\theta$, leading to failure of the authentication system (see Sec. 4.2 below). Hence, the number of images of $\theta$ that the attacker obtains and the size of the authentication sample are of key importance. We now turn to formally stating the problem.

**Notation.** The set of possible observations is denoted by $\mathcal{X}$. Let $\mathcal{H}$ denote the known set of possible sources $\theta$, where each source $\theta \in \mathcal{H}$ is defined by a probability density $f_\theta$, and an observation of a source $\theta \in \mathcal{H}$ is an $\mathcal{X}$-valued random variable with density $f_\theta$. We assume that subsequent observations of the source are IID, so that $n$ sequential observations have density $f_\theta^{(n)}(x_1, \ldots, x_n) := \prod_{i=1}^n f_\theta(x_i)$. We allow $\theta$ to be sampled from a known distribution $Q$ on $\mathcal{H}$ and denote the corresponding $\mathcal{H}$-valued random variable by $\Theta$. In what follows we will denote the number of observations leaked to the attacker by $m$, the number of "registration" observations available to the authenticator by $k$, and the number of observations required at authentication by $n$ (these may be generated by either the attacker or the true source).

**The Authentication Game.** The game begins with a random source $\Theta$ being drawn from $\mathcal{H}$ according to $Q$. The authenticator first receives information about the drawn source, and then chooses a decision rule for deciding whether a given test sequence $x \in \mathcal{X}^n$ is an authentic sequence of observations sampled from $f_\theta^{(n)}$ or a fake sequence generated by the attacker. Formally, the authenticator learns about the source by seeing $k$ IID "registration" observations $A = A_1, \ldots, A_k \sim f_\theta^{(k)}$. The set of all possible decision rules is then $\mathcal{D} : \mathcal{X}^k \times \mathcal{X}^n \to \{0, 1\}$ (where a decision of 1 corresponds to the true source and 0 to an attacker). After the authenticator fixes its strategy, the attacker can seek the best attack strategy. We assume that the attacker has access to $m$ "leaked" IID observations $Y = Y_1, \ldots, Y_m \sim f_\theta^{(m)}$ as information about the source $\theta$. Then it generates an attack sequence $X \in \mathcal{X}^n$ and presents it to the authenticator, which uses its decision rule to decide whether $X$ is an authentic sequence of observations sampled from $f_\theta^{(n)}$ or a fake sequence generated by the attacker. Formally, the strategy set of the attacker is all functions $\mathcal{G} : \mathcal{X}^m \to \Delta(\mathcal{X}^n)$, where $\Delta(\mathcal{X}^n)$ is the set of probability distributions over $\mathcal{X}^n$, and $g_{X|Y}$ is the associated conditional probability density. We note that the set $\mathcal{H}$, the parametric family $f_\theta$, and the prior probability $Q$ are known to both players. Also, note that the leaked sample $Y$ revealed to the attacker is *not* available to the authenticator, and the "registration" sample $A$ is not available to the attacker.

The goal of the authenticator is to maximize its expected accuracy, and the goal of the attacker is to minimize it (or equivalently maximize its success probability). We define the utility (payoff) of

the game as the expected prediction accuracy of the authenticator. To define expected accuracy we consider the case of equal priors for attack and real samples.[1] Formally, for a pair of strategies $(\mathcal{D}, \mathcal{G})$ and a specific source $\theta$, the expected accuracy of the authenticator is then:

$$V(\theta, \mathcal{D}, \mathcal{G}) = \frac{1}{2}\mathbb{E}_{A \sim f_\theta^{(k)}}\mathbb{E}_{Y \sim f_\theta^{(m)}}\left[\mathbb{E}_{X \sim f_\theta^{(n)}}[\mathcal{D}(A, X)] + \mathbb{E}_{X \sim \mathcal{G}(Y)}[1 - \mathcal{D}(A, X)]\right] \quad (2.1)$$

Since this utility only depends on $\mathcal{G}$ in the second term, minimizing it is equivalent to $\mathcal{G}$ maximizing its success probability. To obtain the overall utility for the authenticator, we take the expected value w.r.t $\Theta$ and define $V(\mathcal{D}, \mathcal{G}) = \mathbb{E}_{\Theta \sim Q}V(\Theta, \mathcal{D}, \mathcal{G})$. Finally, we arrive at the following maximin game:

$$V_{game} = \max_{\mathcal{D} \in \mathbb{D}} \min_{\mathcal{G} \in \mathbb{G}} V(\mathcal{D}, \mathcal{G}) \quad (2.2)$$

where $\mathbb{D}, \mathbb{G}$ are the sets of all possible authenticator and attacker strategies, respectively. In Sec. 4 we show that this game has a Nash equilibrium, we characterize the optimal strategies and game value in general, and find them in closed form for the case of Multivariate Gaussian sources.

## 3 RELATED WORK

**Adversarial hypothesis testing (AHT):** Hypothesis testing (HT) is a rich field in statistics that studies how one can detect whether a sample was generated by one of two sets of distributions. A variant of HT that is related to our work but distinct from it is AHT (e.g., see Brandão et al., 2014; Barni & Tondi, 2013b;a; 2014; Bao et al., 2011; Zhou et al., 2019; Brückner & Scheffer, 2011; Brückner et al., 2012). These works describe an HT setting where the sample is generated by one of two hypotheses classes, but is then modified by an adversary in some restricted way. E.g., in Barni & Tondi (2013b;a; 2014) the adversary can change the sample of one class up to a fixed distance (e.g., Hamming). Given the quality of current generative models and the rapid pace of progress, when considering an impersonation attack, it seems that the only relevant restriction one can assume on an attacker is on the information it has. This is not captured by prior work since it assumes that the adversary has a restricted strategy set. In contrast, our work considers a novel problem setting where both players are not limited in their strategy set in any way. This leads to a novel analysis that focuses on the dependence on the finite information available to each player $(m, n, k)$.

**Adversarial Examples:** It has been observed (Goodfellow et al., 2014b) that deep learning models can be very sensitive to small changes in their input. Such "misleading" inputs are known as *adversarial examples* and much recent work has analyzed the phenomenon (Ilyas et al., 2019; Shamir et al., 2019; Zhang et al., 2019), addressed the problem of robustness to these (Moosavi-Dezfooli et al., 2016; Papernot et al., 2017; Yuan et al., 2017), and studied when robusntess can be certified (Raghunathan et al., 2018; Wong et al., 2018). The setting of robust classification in the presence of adversarial examples can also be thought of as a specific case of AHT (see above), where a classifier is required to predict the class of an observation that could have been perturbed by a restricted adversary (Wong et al., 2018; 2019) or generated by an adversary limited to generating examples that will be classified correctly by humans (Song et al., 2018). In contrast, in our setting the attacker is not limited in any way, nor does it have another utility in addition to impersonating the source. Furthermore, in adversarial examples, there is no notion of limited information for the adversary, whereas our work focuses on the dependence of the game on the information available to the players (sample sizes $n, m, k$).

**GAN:** The GAN model is a game between a generator and a discriminator. While our concept of generative attacks is inspired by GAN, it is very different from it: a successful GAN generator is not necessarily a successful attacker in our setting, and vice-versa (e.g., given sufficiently expressive generators and discriminators, GANs can "memorize" the training data, and thus the discriminator will perform at chance level.[2] Such a discriminator will not be useful as a defense against generative attacks). Unlike GANs, in our setting, sample sizes are of key importance. Thus, our attacker will not memorize the data it sees, as this will be detected when generating $n > m$ examples.

**Conditional GANs:** In conditional GANs (Mirza & Osindero, 2014) the generator uses side information for generating new samples. The attacker in our approach (analogous to GAN generator) has input, but this input is not available to the authenticator (analogous to GAN discriminator). Thus,

---

[1]We give equal prior probability to attack and real samples for simplicity and clarity, and since this is common in GAN formulations. All results can be trivially generalized to any prior probability for attack.

[2]If the generator is not expressive enough, the learned GAN may have small support (Arora et al., 2017).

the objective of the learning process is fundamentally different.

**Few-shot learning and generation:** Our work relates to few-shot learning (Snell et al., 2017; Vinyals et al., 2016; Finn et al., 2017; Lake et al., 2011; Koch et al., 2015) and few-shot generative models (Rezende et al., 2016; Zakharov et al., 2019; Lake et al., 2015; Edwards & Storkey, 2017; Hewitt et al., 2018) in the sense that both authenticator and attacker need to learn from a limited set of observations. However, in our setting the authenticator is required to predict whether a sample came from the true source or an attacker impersonating the source while taking into consideration the amount of information both players have. These are notions that are not part of the general few-shot learning setup. Also, in prior work on few-shot generation, the generator is either measured through human evaluation (Lake et al., 2015) or trained to maximize the likelihood of its generated sample (Rezende et al., 2016; Edwards & Storkey, 2017; Hewitt et al., 2018). In contrast, in our setting the attacker's objective is to maximize the probability that its generated sample will be labeled as real by an authenticator. To this end, we show that the attacker must consider the sample sizes $m, n, k$, which the generative models in prior work do not account for. Furthermore, we show in Sec. F.4, and in Figures 1c,6, that the maximum likelihood (ML) solution is indeed sub-optimal in our setting.

**Image to image translation:** Several GAN models have been introduced for mapping between two domains (Zhu et al., 2017; Huang et al., 2018; Isola et al., 2017; Wang et al., 2017; Park et al., 2019). This relates to our work since the attacker also needs to learn to map the leaked sample to an attack sample. However, in our setting the mapping is not to a different domain but rather to other images from the same distribution, which results in a different objective.

**Data Augmentation:** Generative models have also been used for augmenting data in supervised learning, and in particular few-shot learning (Koch et al., 2015; Snell et al., 2017; Vinyals et al., 2016; Lake et al., 2011). One such approach is Data Augmentation GAN (DAGAN) (Antoniou et al., 2018), which takes as input an image and generates a new image. It relates to our framework in the limited case of $m = 1, n = 2, k = 1$, in the sense that the generator's objective is to map one image to two. However, in DAGAN the only goal of the discriminator is to improve the generator, and the generator is limited to the functional form of adding a new image to the existing one, which is a sub-optimal attack strategy, as can be seen from the Gaussian case of our problem.

## 4 THEORETICAL RESULTS

In this section, we study the game defined in Eq. 2.2. First, in Sec. 4.1 we show the existence of a Nash equilibrium and characterize the optimal strategies for both players. Specifically, we show that the optimal attacker strategy minimizes a certain divergence between the source and the attacker's conditional distribution of $X$ given $A$. Next, Sec. 4.2 shows that when there are more leaked samples $m$ than generated samples $n$, the authenticator will fail. Finally, in Sec. 4.3 we provide a closed-form solution for both attacker and authenticator for the case of multivariate Gaussian distributions and analyze the effect of the dimension of the observations and the sample sizes $m, n$, and $k$. Proofs are provided in the appendix.

### 4.1 CHARACTERIZING THE OPTIMAL STRATEGIES

We begin by showing that the game defined in Eq. 2.2 admits a Nash equilibrium. Namely, Theorem 4.1 below shows that there exists a pair of strategies $(\mathcal{D}^*, \mathcal{G}^*)$ that satisfy:

$$\max_{\mathcal{D} \in \mathbb{D}} \min_{\mathcal{G} \in \mathbb{G}} V(\mathcal{D}, \mathcal{G}) = \min_{\mathcal{G} \in \mathbb{G}} \max_{\mathcal{D} \in \mathbb{D}} V(\mathcal{D}, \mathcal{G}) = V(\mathcal{D}^*, \mathcal{G}^*) \tag{4.1}$$

**Theorem 4.1.** *Consider the attacker $\mathcal{G}^*$ defined by:*

$$g_{X|Y}^* \in \operatorname*{argmin}_{g_{X|Y}} \mathbb{E}_{A \sim f_A} \left[ \int_{x \in \mathcal{X}^n} \left| f_{X|A}(x|A) - g_{X|A}(x|A) \right| dx \right] \tag{4.2}$$

*Where, $f_A(a) = \int_{\theta \in \mathcal{H}} Q(\theta) f_\theta^{(k)}(a) d\theta$ is the marginal density of $A$, $Q_{\Theta|A}$ is the posterior over $\mathcal{H}$ given $A$, and $f_{Y|A}(y|a) = \int_{\theta \in \mathcal{H}} f_\theta^{(m)}(y) Q_{\Theta|A}(\theta|a) d\theta$. Also, $f_{X|A}(x|a) = \int_{\theta \in \mathcal{H}} f_\theta^{(n)}(x) Q_{\Theta|A}(\theta|a) d\theta$ and $g_{X|A}(x|a) = \int_{y \in \mathcal{X}^m} g_{X|Y}(x|y) f_{Y|A}(y|a) dy$ are the conditional densities of $X$ given $A$, generated by the source and the attacker respectively. Consider the authenticator defined by $\mathcal{D}^*(a, x) = I \left[ f_{X|A}(x|a) > g_{X|A}^*(x|a) \right]$, where $I$ is the indicator function. Then $(\mathcal{D}^*, \mathcal{G}^*)$ is a solution of Eq. 2.2 that satisfies Eq. 4.1.*

The proof (see Sec. D) follows by first showing that since $\mathcal{D}(a, x) \in \{0, 1\}$, it holds that for any $\mathcal{G}$, the optimal authenticator strategy is a MAP test between the two hypotheses (true source or attacker). We then show that given $\mathcal{D}^*$, the game objective for $\mathcal{G}$ becomes Eq. 4.2. Namely, the optimal attacker minimizes the $\ell_1$ distance over the space $\mathcal{X}^k \times \mathcal{X}^n$ between the true source's conditional distribution of $X$ given $A$, and its own. Therefore, since the proposed $\mathcal{G}^*$ minimizes Eq. 4.2 by definition, it holds that $\min_{\mathcal{G}} V(\mathcal{D}^*, \mathcal{G}) = V(\mathcal{D}^*, \mathcal{G}^*) = \max_{\mathcal{D}} V(\mathcal{D}, \mathcal{G}^*)$ and it follows that $\mathcal{D}^*, \mathcal{G}^*$ satisfy Eq. 4.1.

## 4.2 Replay Attacks: Authentication failure for $n \leqslant m$

When $n \leqslant m$, the attacker generates a number of observations that is at most the number of observations it has seen. Intuitively, an optimal attack in this case, is to simply "replay" a subset of size $n$ from the $m$ observations. This is known as a replay-attack (Syverson, 1994). This subset constitutes an IID sample of length $n$ of the observed source, and is, therefore, a legitimate "fresh" sample. In this case, it seems like the attack cannot be detected by the authenticator. Indeed it is easy to show using Theorem 4.1 that this attack is optimal and therefore for $n \leqslant m$ we have: $\max_{\mathcal{D} \in \mathbb{D}} \min_{\mathcal{G} \in \mathbb{G}} V(\mathcal{D}, \mathcal{G}) = 0.5$ (see Sec. E)

## 4.3 The Gaussian case

We now turn to the case of multivariate Gaussian distributions where we can find the exact form of the attacker and authenticator, providing insight into the general problem. Specifically, we consider the setting where the source distributions are $d$-dimensional multivariate Gaussians with an unknown mean and known covariance, and the prior $Q$ over $\mathcal{H}$ is the improper uniform prior.[3] We assume $n > m$ to keep the problem non-trivial. Let the observations be $d$-dimensional Gaussian vectors with a known covariance matrix $\Sigma \in \mathbb{R}^{d \times d}$ and an unknown mean vector $\theta \in \mathbb{R}^d$. The set of possible sources $\mathcal{H}$ becomes $\mathbb{R}^d$, the Gaussian mean vectors. For any sample of $n$ examples $z \in \mathbb{R}^{n \times d}$, we let $z_i$ denote the $i$'th example, and $\bar{z} = \frac{1}{n} \sum_{i=1}^{n} z_i$ denote the sample mean. Finally, for any $v \in \mathbb{R}^d, B \in \mathbb{R}^{d \times d}$, we define $\|v\|_B^2 = v^T B v$. The following theorem gives a closed-form solution for both attacker and authenticator for the game defined in Eq. 2.2.

**Theorem 4.2.** *Define $\delta = m/n \leqslant 1$ and let $\rho = m/k$. Consider the attacker $\mathcal{G}^*$ defined by the following generative process: Given a leaked sample $Y \in \mathbb{R}^{m \times d}$, $\mathcal{G}^*$ generates a sample $X \in \mathbb{R}^{n \times d}$ as follows: it first samples $n$ vectors $W_1, \ldots, W_n \overset{iid}{\sim} \mathcal{N}(0, \Sigma)$ and then sets: $X_i = W_i - \bar{W} + \bar{Y}$. Define the authenticator $\mathcal{D}^*$ by:*

$$\mathcal{D}^*(a, x) = I\left[ \|\bar{x} - \bar{a}\|_{\Sigma^{-1}}^2 < \frac{d(1+\rho)(1+\rho\delta^{-1})}{n(1-\delta)} \log\left(\frac{\rho+1}{\rho+\delta}\right) \right] \tag{4.3}$$

*Then $(\mathcal{D}^*, \mathcal{G}^*)$ is a solution of Eq. 2.2 that satisfies Eq. 4.1.*

The proof (see Sec. F) starts by showing that $\forall \alpha > 0$, given $\mathcal{D}(a, x) = I[\|\bar{x} - \bar{a}\|_{\Sigma^{-1}}^2 < \alpha]$, the optimal strategy for $\mathcal{G}$ is to set $\bar{x} = \bar{y}$ with probability 1 (as done in $\mathcal{G}^*$). To prove this, we first use the Prekopa-Leindler inequality (Prékopa, 1973) to show that in this case $\mathcal{G}$'s maximization objective is log-concave. We then show that any $\mathcal{G}$ that satisfies $\bar{x} = \bar{y}$ with probability 1 is a local extremum, and since the objective is log-concave it follows that it is the global maximum. We continue by showing that given $\mathcal{G}^*$, the proposed $\mathcal{D}^*$ is optimal. To do so, we first find the distribution of $\mathcal{G}^*$'s attack sample using the Woodbury matrix identity, and then show that $\mathcal{D}^*$ is indeed the optimal decision rule. Finally, using the max-min inequality, this implies that $(\mathcal{D}^*, \mathcal{G}^*)$ satisfy Eq. 4.1.

There are several interesting observations about the above optimal strategies. Perhaps the most intuitive strategy for the attacker would have been to sample $n$ elements from a Gaussian with mean $\bar{Y}$ and the known covariance $\Sigma$. In expectation, this sample would have the correct mean. However, this turns out to be sub-optimal, as can be seen in Figures 1c and 6 (we refer to this as an ML attack. See Sec. F.4 in the appendix for the derivation and visualizations). Instead, the optimal attacker begins by drawing an IID sample $W$ from a Gaussian distribution with mean 0, and then "forces" the sample mean to be exactly $\bar{Y}$ by shifting the sample points by $\bar{Y} - \bar{W}$. This optimal attacker

---

[3]Similar results can be derived for the conjugate prior case, namely, proper Gaussian priors.

strategy can be viewed as matching the sufficient statistics of the leaked sample $Y$ in the generated sample $X$. The optimal authenticator is a MAP test for the optimal attacker, as in Theorem 4.1.

As a corollary to Theorem 4.2 we obtain the value of the game (i.e., the accuracy of $\mathcal{D}^*$).

**Corollary 4.3.** *Define $\delta$ and $\rho$ as in Theorem 4.2. Then the game value for the Gaussian case is:*

$$\frac{1}{2} + \frac{1}{2\Gamma(\frac{d}{2})} \left[ \gamma \left( \frac{d}{2}, \frac{d(1+\rho)}{2(1-\delta)} \log \frac{1+\rho}{\delta+\rho} \right) - \gamma \left( \frac{d}{2}, \frac{d(\delta+\rho)}{2(1-\delta)} \log \frac{1+\rho}{\delta+\rho} \right) \right] \tag{4.4}$$

*Where $\gamma$ is the lower incomplete Gamma function, and $\Gamma$ is the Gamma function.*

The proof (see Sec. F) follows by showing that the test statistic used by $\mathcal{D}^*$ is Gamma distributed. Fig. 1 demonstrates several interesting aspects of the above results. First, Fig. 1a shows that the authenticator accuracy improves as $n$ (the size of the test sample) grows. Furthermore, accuracy also improves as the dimension $d$ grows, meaning that for a specified authentication accuracy, the required ratio $n/m$ becomes smaller with the dimension $d$. This is a very encouraging result since although this dimensional dependency is proved only for Gaussian sources, it suggests that for real-world high-dimensional sources (e.g., faces, video, voice, etc.) the authenticator can achieve high accuracy even when requiring a small (and practical) authentication sample.

Intuitively it may seem like authentication is impossible when the authenticator has less data than the attacker (i.e., $m > k$). Surprisingly, this is not the case. As can be seen in Fig. 1b, even when $m > k$, the authenticator can achieve non trivial accuracy.[4] An intuitive explanation of this phenomenon is that the test statistic used by the authenticator is $\|\bar{X} - \bar{A}\|$, which, due to the variance in the attacker estimation, has higher variance when $X$ is generated by an attacker than when $X$ is generated by the true source. This, in turn, allows the authenticator to discriminate between the hypotheses.

A closed-form solution for the general case remains an open problem. We believe that solving for Gaussians is an important step forward, since it exposes interesting structural properties of the solution, which we use in practice. Furthermore, if $\mathcal{G}$ has an encoder-decoder structure, it is not unreasonable that the source distribution in latent space can be approximately Gaussian (as in VAE).

## 5 GAN IN THE MIDDLE NETWORKS

So far we explored the general formalism of authentication games. Here we consider specific architectures for $\mathcal{D}$ and $\mathcal{G}$. As in GAN based models (Mirza & Osindero, 2014; Mescheder et al., 2018; Karras et al., 2018a;b), we use neural nets to model these, while using insights from our theoretical analysis. Below we provide implementation details for the GIM model (see Sec. H and code for more details). In our analysis, we considered the non-differentiable zero-one loss since it is the real accuracy measure. In practice, we will use cross-entropy as used in most GAN approaches.

**Authenticator Architecture:** The authenticator is implemented as a neural network $\mathcal{D}(a, x)$ that maps from a source information sample $a \in \mathcal{X}^k$ and a test sample $x \in \mathcal{X}^n$ to a probability that the test sample came from the true source. Our framework does not restrict the authenticator to any specific function type, but in practice one must implement it using some model. We recall that our theoretical results do suggest a certain functional form. The Gaussian results in Sec. 4.3 show that the optimal authenticator is a test on the sufficient statistic of the source parameters. Motivated by this result, and in the spirit of Siamese networks (Koch et al., 2015; Chopra et al., 2005), we consider the following form for the authenticator. We define a function $T_{\mathcal{D}}$ that maps a sample to a fixed sized vector, analogous to the sufficient statistic in the theorem. We apply $T_{\mathcal{D}}$ to both $a$ and $x$. Then, these two outputs are used as input to a comparison function $\sigma$ which outputs a scalar reflecting their similarity. Thus the authenticator can be expressed as: $\mathcal{D}(a, x) = \sigma(T_{\mathcal{D}}(a), T_{\mathcal{D}}(x))$.

**Attacker Architecture:** The attacker is implemented as a stochastic neural network $\mathcal{G}(y)$ that maps a leaked sample $y \in \mathcal{X}^m$ to an attack sample $x \in \mathcal{X}^n$. Our theoretical results suggest a certain functional form for this network. The Gaussian analysis in Sec. 4.3 shows that the optimal attacker generates a sample whose sufficient statistic matches that of the leaked sample. Motivated by this result, we consider the following functional form for the attacker. First, it applies a function $T_{\mathcal{G}}$ that maps the leaked sample $Y$ to a fixed sized vector $T_{\mathcal{G}}(Y)$, analogous to the sufficient statistic in the

---

[4] See Sec. C in the appendix, for additional figures showing that as the dimension grows, the expected accuracy goes to 1.

theorem. It then draws $n$ random latent vectors $W_1, \ldots, W_n$ and matches their mean to the leaked sufficient statistic to obtain the latent vectors $W'_i$. Namely it sets $W'_i = W_i - \bar{W} + T_{\mathcal{G}}(Y)$ as done in Theorem 4.2. Finally, it uses a decoder function $\varphi$ that maps each latent vector $W'_i$ to the domain $\mathcal{X}$. Thus, the attacker can be expressed as: $\mathcal{G}(Y)_i = \varphi(W_i - \bar{W} + T_{\mathcal{G}}(Y)) \quad \forall i \in [n]$.

**Optimization Details:** Each iteration begins when a source $\theta$ is chosen randomly from the set of sources in the training dataset (e.g., a person to be authenticated). Samples $A, Y, X_\theta$ are drawn from the set of examples available for $\theta$, where $X_\theta$ represents a test sample from $\theta$. Then, given a leaked sample $Y$, the generator $\mathcal{G}$ generates a fake sample $X_{\mathcal{G}}$, passes it to $\mathcal{D}$ and suffers the appropriate loss. Finally, $\mathcal{D}$ receives as input the source information sample $A$, outputs a prediction for each of the test samples $X_\theta, X_{\mathcal{G}}$, and suffers the appropriate loss. Optimization is done via gradient ascent on authenticator parameters and descent on attacker parameters, as is typical for GAN problems.

# 6 EXPERIMENTS

We next evaluate our method empirically. In all experiments, we use the model described in Sec. 5. We optimize the model with adversarial training, using the loss suggested by Mescheder et al. (2018) and Adam (Kingma & Ba, 2015). Our implementation is available at `https://github.com/roymor1/OptimalStrategiesAgainstGenerativeAttacks`. Also, see Sec. H for further implementation details.

**Gaussian sources:** For the case of Gaussian sources, we arrived at a complete characterization of the solution in Sec. 4.3. Thus, we can learn the models using our GIM algorithm and check whether it finds the correct solution. This is important, as the GIM objective is clearly non-convex and GANs are generally hard to train in practice and lack convergence guarantees (Mescheder et al., 2018). We ran all experiments using a multivariate Gaussian with $\Sigma = I_d$, and in each game the source mean was drawn from the prior distribution $Q = \mathcal{N}(0, 10I_d)$. This approximates the improper uniform prior since the prior has much larger variance than the sources. Fig. 1a shows the empirical game value compared with the theoretical one as a function of the test sample size $n$, for fixed $m = 1, k = 10$, and different values of $d$. It can be seen that there is an excellent fit between theory and experiment.

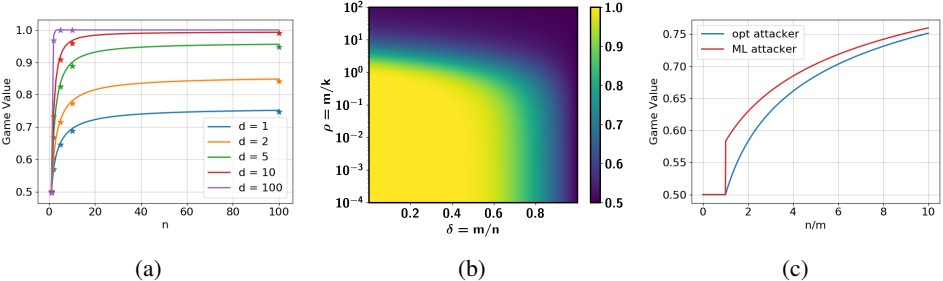

(a)               (b)               (c)

Figure 1: Game value (expected authentication accuracy) for the Gaussian case. (a) A comparison between empirical and theoretical game value for different $d$ values ($m = 1, k = 10$). Solid lines describe the theoretical game values whereas the $*$ markers describe the empirical accuracy when learning with the GIM model. (b) Theoretical game value as a function of $\delta, \rho$ (see Corollary 4.3) for $d = 100$. (c) Empirical accuracy of an optimal authenticator against two attacks: the theoretically optimal attack $\mathcal{G}^*$ from Theorem 4.2 and a maximum likelihood (ML) attack (See Sec. F.4) for the Gaussian case. It can be seen that the ML attack is inferior in that it results in better accuracy for the authenticator, as predicted by our theoretical results.

**Authentication on Faces and Characters:** We next evaluate GIM in an authentication setting on two datasets: the VoxCeleb2 faces dataset (Nagrani et al., 2017; Chung & Zisserman, 2018), and the Omniglot handwritten character dataset (Lake et al., 2015). Additional information about the datasets, splits, and modeling details is provided in Sections G and H. Our goal is to check whether the GIM authenticator is more robust to generative attacks than a state of the art authentication system. To evaluate this, we consider several attackers: 1) A "random source" attacker (RS): a naive attacker that ignores the leaked sample $Y$. It simply draws a random source from the dataset, and samples $n$ real images of that source. From the authenticator's perspective, it's equivalent to a sample version of the verification task (Koch et al., 2015; Schroff et al., 2015; Deng et al., 2018), in

Table 1: Accuracy of GIM and baselines against attacks. Avg acc denotes average over all attacks.

| Authenticator | Dataset | m | n | k | RS | Replay | GIM | Avg acc |
|---|---|---|---|---|---|---|---|---|
| GIM | VoxCeleb2 | 1 | 5 | 5 | 0.897 | 0.837 | 0.822 | **0.852** |
| ArcFace | VoxCeleb2 | 1 | 5 | 5 | 0.998 | 0.598 | 0.526 | 0.707 |
| GIM | Omniglot | 1 | 5 | 5 | 0.912 | 0.942 | 0.868 | **0.907** |
| Siamese | Omniglot | 1 | 5 | 5 | 0.994 | 0.509 | 0.785 | 0.763 |

which an agent is presented with a pair of real images and needs to decide whether they are from the same source or not. 2) Replay attacker (Replay): an attacker which, upon seeing a leaked sample $Y$, draws $n$ random images of the leaked sample (with replacement). 3) A GIM attack, which is the "worst case" attacker $\mathcal{G}$, learned by our GIM model.

For VoxCeleb we compare the GIM authenticator to the ArcFace method (Deng et al., 2018), which is currently state of the art in face verification. As a baseline for Omniglot, we use the Siamese network suggested by Koch et al. (2015), which achieves state of the art in the verification task on Omniglot. Results are shown in Table 1. It can be seen that on average across attacks, GIM outperforms the baselines. The only attack for which GIM is inferior is RS. This is not surprising as this is the objective that both baselines are trained for.

**Qualitative evaluation of attacker:** In Fig. 2 we provide images generated by the GIM attacker for the test set of both Omniglot and Voxceleb. The images demonstrate qualitatively the strategy learned by the attacker. In the Voxceleb2 dataset, face images are drawn from a video of the person talking. Note that as in real samples from the data, the attack sample varies in pose and expression and not in the background or clothing.

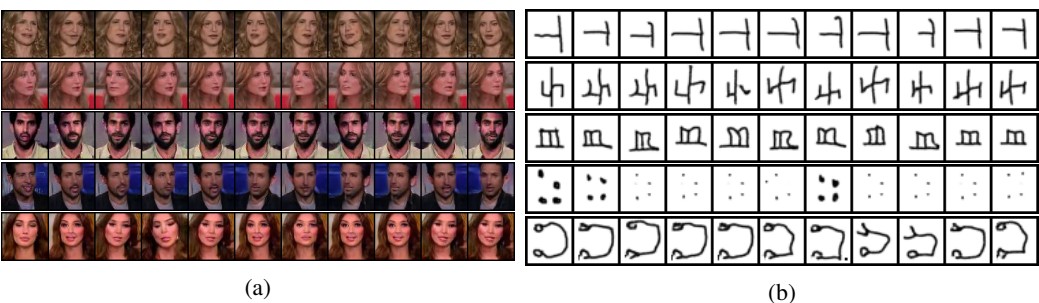

(a)  (b)

Figure 2: Images generated by the GIM attacker based on one leaked image. In each row, the leftmost image is the real leaked image, and the rest of the images are an attack sample generated by the GIM attacker. (a) Voxceleb2 dataset. (b) Omniglot dataset.

**Data augmentation:** Finally, we use GIM for data augmentation in one-shot classification on Omniglot. This is done by using the GIM attacker to generate more data for a given class. We first train GIM on the training set with parameters $m = 1, n = 5, k = 5$. Then, during both training and testing of one-shot classification, given an example, we use the attacker to augment the single example available for each of the classes, by adding to it the $n = 5$ examples our attacker generated from it. We use Prototypical Nets (Snell et al., 2017) as the baseline model. We find that without using our augmentation method, Prototypical Nets achieve 95.9% accuracy on the test split, and with our method, they achieve 96.5%, which is similar to the improvement achieved by Antoniou et al. (2018) with Matching networks (Vinyals et al., 2016) as the few-shot classification algorithm.

## 7  CONCLUSIONS

We defined the notion of authentication in the face of generative attacks, in which a generative model attempts to produce a fake reality based on observations of reality. These attacks raise numerous interesting theoretical questions and are very important and timely from a practical standpoint. We proposed to study generative attacks as a two-person zero-sum game between attacker and authenticator. In our most general setup both attacker and authenticator have access to a finite set of

observations of the source. We show that this game has a Nash equilibrium, and we characterize the optimal strategies. In the Gaussian version of the game, a closed form of the optimal strategies is available. A nice outcome of the analysis is that the game value depends on $m, n, k$ only through their ratios $\delta = m/n$ (i.e., the "expansion ratio" between attack and leaked sample sizes) and $\rho = m/k$ (i.e., the "information ratio" between the number of source observations available to attacker and authenticator). As we show in Fig. 1b, there is a large range of values for which high accuracy authentication is possible, and as $d$ grows we observe that the high authentication accuracy region in the $(\delta, \rho)$ plane grows sharply. We introduce the GIM model, which is a practical approach to learning both authenticator and attacker, and whose structure is inspired by our analysis. GIM achieves accuracy that is very close to the theoretical rates in the Gaussian case, and is also more robust to attacks when compared to state of the art authenticators on real data. Many theoretical and practical questions remain. For example, finding closed form optimal strategies for other distributions, and going beyond IID generation. The non IID setting is of particular importance for the problem of fake video (Thies et al., 2016) and audio (Arik et al., 2018) generation, which we intend to study in the future.

## ACKNOWLEDGEMENTS

This work has been supported by the Blavatnik Interdisciplinary Research Center (ICRC), the Federmann Research Center (Hebrew University) and Israeli Science Foundation research grants 1523/16 and 1186/18.

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

## A  INTRODUCTION TO THE APPENDIX

Here we provide additional information and proofs for the main paper. In Sec. B we provide a visualization of the game setting. In Sec. C we plot the game value in the Gaussian case for different parameter values. In Sec. D we provide a proof for Theorem 4.1, in Sec. E we formally state the theorem discussed in Sec. 4.2 and prove it, and in Sec. F we prove Theorem 4.2 and also derive the game value for the sub-optimal "ML attacker". Finally, in Sections G, H we provide additional details about the experiments and the implementation of GIM.

## B  PROBLEM SETUP VISUALIZATION

Our problem setup is illustrated in Fig. 3 for a face authentication scenario. A source $\theta$ (in this case an individual) generates IID observations (images). $k$ images are used by the authenticator to study the source which it aims to authenticate. $m$ images are "leaked" and obtained by an attacker who wishes to impersonate the source and pass the authentication. At test time the authenticator is presented with $n$ images that were generated either by the true source $\theta$, or by the attacker, and decides whether the entity that generated the images was the source $\theta$ or an attacker.

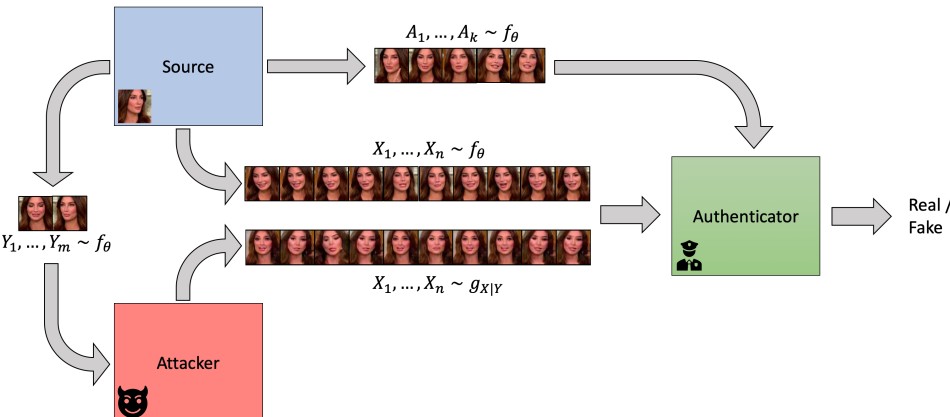

Figure 3: An illustration of the game described in the main text. An authenticator receives a sample of $n$ images and needs to decide whether these were generated by a known source, or by an adversary that had access to leaked images. In order to decide, the authenticator is supplied with a sample of $k$ images from the real source.

## C  GAME VALUE VISUALIZATIONS

In corollary 4.3 we provide the game value (the expected authenticator accuracy) for the case of multivariate Gaussian sources. In this section, we present additional visualizations of the game value as a function of the different parameters of the game. Fig. 4 visualizes the game value as a function of $\delta$ and $\rho$, for different values of $d$ (the observation dimension). $\delta = \frac{m}{n}$ is the "expansion ratio" between the source information available to the attacker through the leaked sample and the size of the attacker's attack sample. $\rho = \frac{m}{k}$ is the "information ratio" between the number of source observations available to attacker and authenticator. One can clearly see that as $d$, the observation dimension, grows large, so does the accuracy of the authenticator. Even for $\delta$ values higher than $0.5$, and $\rho$ values which intuitively would give the attacker an advantage (e.g., $\frac{m}{k} = 10$).

Fig. 5 visualizes the game value as a function of $\delta$ and $\epsilon$, for different values of $d$ (the observation dimension). Where $\epsilon = \frac{n}{k}$ is the "information expansion" of the attacker with respect to the authenticator source information. Again, one can clearly see that as $d$, the observation dimension, grows large, so does the accuracy of the authenticator.

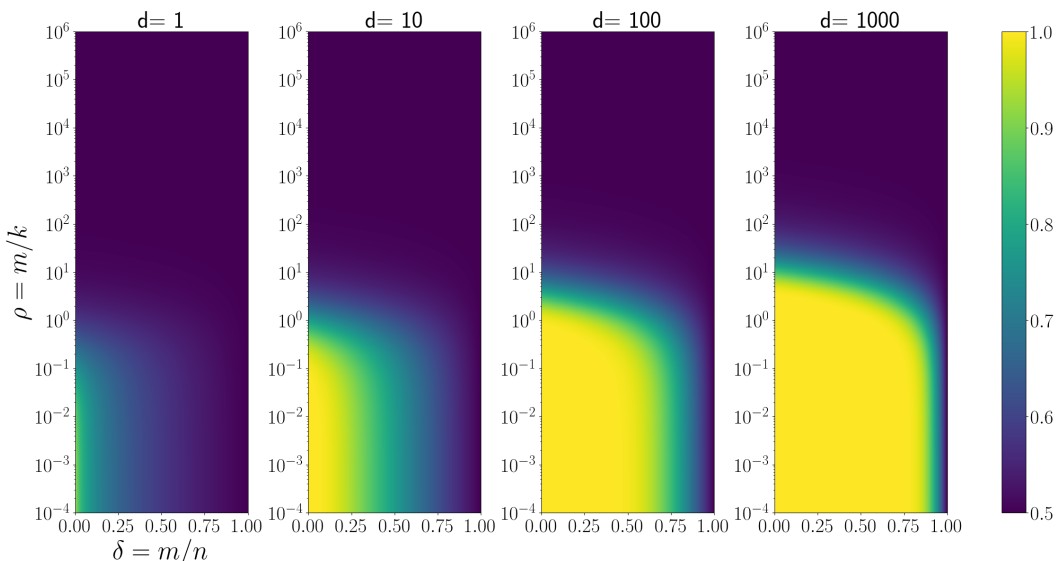

Figure 4: Game value as a function of $\delta, \rho$ for different dimensions $d$.

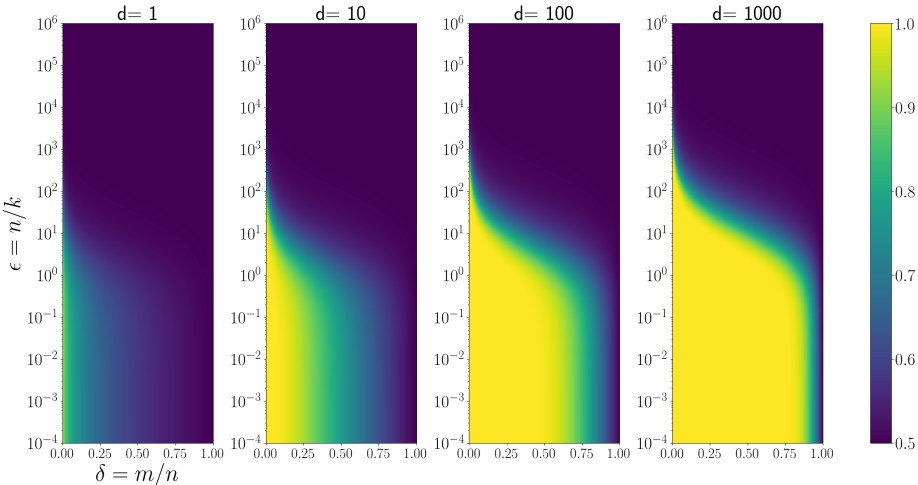

Figure 5: Game value as a function of $\delta, \epsilon$ for different dimensions $d$.

# D   ADDITIONAL THEOREMS AND PROOFS FOR SEC. 4.1

We begin with some additional notation. Let

$$f_A(a) = \int_{\theta \in \mathcal{H}} Q(\theta) f_\theta^{(k)}(a) d\theta$$

denote the marginal density of $A$. Let $Q_{\Theta|A}$ denote the posterior probability over $\mathcal{H}$ given $A$. That is:

$$Q_{\Theta|A}(\theta|a) = \frac{Q(\theta)f_\theta^{(k)}(a)}{\int_{\nu \in \mathcal{H}} Q(\nu)f_\nu^{(k)}(a)d\nu} \equiv \frac{Q(\theta)f_\theta^{(k)}(a)}{f_A(a)}$$

Also, let $f_{Y|A}, f_{X|A}, g_{X|A}$, denote the conditional densities defined by:

$$f_{Y|A}(y|a) = \int_{\theta \in \mathcal{H}} f_\theta^{(m)}(y)Q_{\Theta|A}(\theta|a)d\theta$$

$$f_{X|A}(x|a) = \int_{\theta \in \mathcal{H}} f_\theta^{(n)}(x)Q_{\Theta|A}(\theta|a)d\theta$$

$$g_{X|A}(x|a) = \int_{y \in \mathcal{X}^m} g_{X|Y}(x|y)f_{Y|A}(y|a)dy$$

**Lemma D.1.** *Let $\mathcal{G}$ be an attacker defined by the conditional probability distribution $g_{X|Y}$. Then $\forall a \in \mathcal{X}^k, x \in \mathcal{X}^n$ a best response strategy for $\mathcal{D}$ is:*

$$\mathcal{D}(a,x) = I[f_{X|A}(x|a) > g_{X|A}(x|a)] \tag{D.1}$$

*Proof.* Given an attacker strategy $g_{X|Y}$, the objective for $\mathcal{D}$ is given by:

$$\underset{\mathcal{D}}{\operatorname{argmax}} \, \mathbb{E}_{\Theta \sim Q} V(\Theta, \mathcal{D}, \mathcal{G})$$

$$= \underset{\mathcal{D}}{\operatorname{argmax}} \, \mathbb{E}_{\Theta \sim Q} \mathbb{E}_{A \sim f_\Theta^{(k)}} \left[ \mathbb{E}_{X \sim f_\Theta^{(n)}}[\mathcal{D}(A,X)] + \mathbb{E}_{Y \sim f_\Theta^{(m)}} \mathbb{E}_{X \sim g_{X|Y}(\cdot|Y)}[1 - \mathcal{D}(A,X)] \right]$$

$$= \underset{\mathcal{D}}{\operatorname{argmax}} \, \mathbb{E}_{A \sim f_A} \left[ \mathbb{E}_{X \sim f_{X|A}(\cdot|A)}[\mathcal{D}(A,X)] + \mathbb{E}_{Y \sim f_{Y|A}(\cdot|A)} \mathbb{E}_{X \sim g_{X|Y}(\cdot|Y)}[1 - \mathcal{D}(A,X)] \right]$$

$$= \underset{\mathcal{D}}{\operatorname{argmax}} \int_{a \in \mathcal{X}^k} f_A(a) \int_{x \in \mathcal{X}^n} \left[ f_{X|A}(x|a)\mathcal{D}(a,x) + g_{X|A}(x|a)[1 - \mathcal{D}(a,x)] \right] dx da$$

Note that $\mathcal{D}$ can be optimized independently for each pair $a, x \in \mathcal{X}^k \times \mathcal{X}^n$. Hence, $\forall a, x \in \mathcal{X}^k \times \mathcal{X}^n$ the objective is:

$$\underset{\mathcal{D}(a,x) \in \{0,1\}}{\operatorname{argmax}} \left\{ f_{X|A}(x|a)\mathcal{D}(a,x) + g_{X|A}(x|a)[1 - \mathcal{D}(a,x)] \right\}$$

And thus, the optimal decision rule for $\mathcal{D}$ is:

$$\mathcal{D}(a,x) = I\left[ f_{X|A}(x|a) > g_{X|A}(x|a) \right]$$

As required. $\qquad\qquad\qquad\qquad\qquad\qquad\qquad\qquad\qquad\qquad\qquad\qquad\qquad\qquad\qquad\qquad\qquad\quad\square$

**Lemma D.2.** *$\forall a, x \in \mathcal{X}^k \times \mathcal{X}^n$ let the strategy for $\mathcal{D}$ be defined by:*

$$\mathcal{D}(a,x) = I\left[ f_{X|A}(x|a) > g_{X|A}(x|a) \right]$$

*Then $\mathcal{G}^*$ is a best response strategy for $\mathcal{G}$, iff it minimizes the $\ell_1$ distance between the distributions $f_{X|A}, g_{X|A}$ over the space $\mathcal{X}^k \times \mathcal{X}^n$. Namely:*

$$g_{X|Y}^* \in \underset{g_{X|Y}}{\operatorname{argmin}} \, \mathbb{E}_{A \sim f_A} \int_{x \in \mathcal{X}^n} \left| f_{X|A}(x|A) - g_{X|A}(x|A) \right| dx \tag{D.2}$$

*Proof.* Let $\mathcal{D}$ be defined as in Eq. D.1, the objective for $\mathcal{G}$ is:

$$\underset{\mathcal{G}}{\arg\min} \frac{1}{2}\mathbb{E}_{\Theta\sim Q}V(\Theta,\mathcal{D},\mathcal{G})$$

$$= \underset{g_{X|Y}}{\arg\min} \mathbb{E}_{\Theta\sim Q}\mathbb{E}_{A\sim f_{\Theta}^{(k)}} \left[ \mathbb{E}_{X\sim f_{\Theta}^{(n)}}[\mathcal{D}(A,X)] + \mathbb{E}_{Y\sim f_{\Theta}^{(m)}}\mathbb{E}_{X\sim g_{X|Y}(\cdot|Y)}[1-\mathcal{D}(A,X)] \right]$$

$$= \underset{g_{X|Y}}{\arg\min} \mathbb{E}_{A\sim f_A} \left[ \mathbb{E}_{X\sim f_{X|A}(\cdot|A)}[\mathcal{D}(A,X)] + \mathbb{E}_{X\sim g_{X|A}(\cdot|A)}[1-\mathcal{D}(A,X)] \right]$$

$$= \underset{g_{X|Y}}{\arg\min} \int_{a\in\mathcal{X}^k} da\, f_A(a) \int_{x\in\mathcal{X}^n} dx \left[ f_{X|A}(x|a)\mathcal{D}(a,x) + g_{X|A}(x|a)[1-\mathcal{D}(a,x)] \right]$$

$$= \underset{g_{X|Y}}{\arg\min} \int_{a\in\mathcal{X}^k} da\, f_A(a) \int_{x\in\mathcal{X}^n} dx \max\left\{ f_{X|A}(x|a), g_{X|A}(x|a) \right\} \tag{D.3}$$

$$= \underset{g_{X|Y}}{\arg\min} \frac{1}{2} \int_{a\in\mathcal{X}^k} da\, f_A(a) \int_{x\in\mathcal{X}^n} dx \left[ f_{X|A}(x|a) + g_{X|A}(x|a) + |f_{X|A}(x|a) - g_{X|A}(x|a)| \right]$$

$$= \underset{g_{X|Y}}{\arg\min}\, 1 + \frac{1}{2} \int_{a\in\mathcal{X}^k} da\, f_A(a) \int_{x\in\mathcal{X}^n} dx\, |f_{X|A}(x|a) - g_{X|A}(x|a)|$$

$$= \underset{g_{X|Y}}{\arg\min} \int_{a\in\mathcal{X}^k} da\, f_A(a) \int_{x\in\mathcal{X}^n} dx\, |f_{X|A}(x|a) - g_{X|A}(x|a)|$$

As required. Where in Eq. D.3 we used the definition of $\mathcal{D}$. $\square$

**Theorem 4.1.** *Consider the attacker defined by:*

$$g_{X|Y}^* \in \underset{g_{X|Y}}{\arg\min}\, \mathbb{E}_{A\sim f_A} \left[ \int_{x\in\mathcal{X}^n} |f_{X|A}(x|A) - g_{X|A}(x|A)|\, dx \right] \tag{D.4}$$

*and let $\mathcal{G}^*$ be the corresponding map from $\mathcal{X}^m$ to the set of probability distributions over $\mathcal{X}^n$. Consider the authenticator defined by:*

$$\mathcal{D}^*(a,x) = I\left[ f_{X|A}(x|a) > g_{X|A}^*(x|a) \right] \tag{D.5}$$

*where $I$ is the indicator function. Then $(\mathcal{D}^*, \mathcal{G}^*)$ is a solution of Eq. 2.2 that satisfies Eq. 4.1.*

*Proof.* From Lemmas D.1, D.2 we have that $\max_{\mathcal{D}} V(\mathcal{D},\mathcal{G}^*) = V(\mathcal{D}^*,\mathcal{G}^*) = \min_{\mathcal{G}} V(\mathcal{D}^*,\mathcal{G})$, from which it follows that Eq. 4.1 is satisfied and thus $(\mathcal{D}^*, \mathcal{G}^*)$ is a solution of Eq. 2.2. $\square$

## E  THEOREM AND PROOF FOR SEC. 4.2

**Theorem E.1.** *For all $n \leqslant m$ it holds that:*

$$\max_{\mathcal{D}\in\mathbb{D}} \min_{\mathcal{G}\in\mathbb{G}} V(\mathcal{D},\mathcal{G}) = 0.5$$

*Proof.* Consider the attacker $\mathcal{G}_{\text{replay}}$ defined by the following generative process: Given a leaked sample $Y \in \mathbb{R}^{m\times d}$, $\mathcal{G}_{\text{replay}}$ generates a sample $X \in \mathbb{R}^{n\times d}$ such that $X_i = Y_i \quad \forall i \in [n]$. (this is possible since we assumed $n \leqslant m$). Namely, we have:

$$g_{X|Y}(x|y) = \prod_{i=1}^{n} \delta(x_i - y_i)$$

Where $\delta$ is the Dirac delta. Thus $\forall a, x \in \mathcal{X}^k \times \mathcal{X}^n$:

$$
\begin{aligned}
g_{X|A}(x|a) &= \int_{y \in \mathcal{X}^m} dy\, g_{X|Y}(x|y) f_{Y|A}(y|a) \\
&= \int_{y \in \mathcal{X}^m} dy\, g_{X|Y}(x|y) \int_{\theta \in \mathcal{H}} d\theta\, f_\theta^{(m)}(y) Q_{\Theta|A}(\theta|a) \\
&= \int_{\theta \in \mathcal{H}} d\theta \int_{y \in \mathcal{X}^m} dy \prod_{i=1}^{n} \delta(x_i - y_i) f_\theta^{(m)}(y) Q_{\Theta|A}(\theta|a) \\
&= \int_{\theta \in \mathcal{H}} d\theta\, Q_{\Theta|A}(\theta|a) \int_{y \in \mathcal{X}^n} dy \prod_{i=1}^{n} \delta(x_i - y_i) f_\theta^{(n)}(y) \int_{y' \in \mathcal{X}^{m-n}} dy'\, f_\theta^{(m-n)}(y') \\
&= \int_{\theta \in \mathcal{H}} d\theta\, Q_{\Theta|A}(\theta|a) \int_{y \in \mathcal{X}^n} dy \prod_{i=1}^{n} \delta(x_i - y_i) f_\theta^{(n)}(y) \\
&= \int_{\theta \in \mathcal{H}} d\theta\, Q_{\Theta|A}(\theta|a) f_\theta^{(n)}(x) \\
&= f_{X|A}(x|a)
\end{aligned}
$$

Define:

$$
\mathcal{D}_0(a, x) = 0 \quad \forall a, x \in \mathcal{X}^k \times \mathcal{X}^n
$$

Then according to Theorem 4.1 $(\mathcal{D}_0, \mathcal{G}_{\text{replay}})$ is a solution of Eq. 2.2 that satisfies Eq. 4.1, and therefore:

$$
\begin{aligned}
\max_{\mathcal{D} \in \mathbb{D}} \min_{\mathcal{G} \in \mathbb{G}} V(\mathcal{D}, \mathcal{G}) &= V(\mathcal{D}_0, \mathcal{G}_{\text{replay}}) \\
&= \frac{1}{2} \mathbb{E}_{\Theta \sim Q} \mathbb{E}_{A \sim f_\theta^{(k)}} \mathbb{E}_{Y \sim f_\theta^{(m)}} \left[ \mathbb{E}_{X \sim f_\theta^{(n)}}[\mathcal{D}(A, X)] + \mathbb{E}_{X \sim \mathcal{G}(Y)}[1 - \mathcal{D}(A, X)] \right] \\
&= \frac{1}{2} \mathbb{E}_{\Theta \sim Q} \mathbb{E}_{A \sim f_\theta^{(k)}} \mathbb{E}_{Y \sim f_\theta^{(m)}} \left[ \mathbb{E}_{X \sim f_\theta^{(n)}}[0] + \mathbb{E}_{X \sim \mathcal{G}(Y)}[1] \right] \\
&= \frac{1}{2}
\end{aligned}
$$

As required. $\qquad\square$

## F  ADDITIONAL THEOREMS AND PROOFS FOR SEC. 4.3

### F.1  NOTATION AND DEFINITIONS

In this section, we consider the case where the sources are $d$-dimensional Gaussian vectors with a known covariance matrix $\Sigma = CC^T \in \mathbb{R}^{d \times d}$ and unknown mean vector $\theta \in \mathbb{R}^d$. That is, the set of possible sources is $\mathcal{H} = \mathbb{R}^d$ and given $\theta \in \mathcal{H}$ the associated probability density over the domain $\mathcal{X} = \mathbb{R}^d$ is $f_\theta(x) = \frac{1}{\sqrt{(2\pi)^d |\det(\Sigma)|}} \exp\left(-\frac{1}{2}(x - \theta)^T \Sigma^{-1}(x - \theta)\right)$

A sample of $n$ examples $x \in \mathcal{X}^n$ is considered a matrix $x \in \mathbb{R}^{n \times d}$, where the first index represents the sample and the second represents the observation space, $\mathbb{R}^d$. I.e., $x_{ij} \in \mathbb{R}$ is the $j$'th element of the $i$'th example in the sample $x \in \mathbb{R}^{n \times d}$.

We continue with a few more notations that simplify the proofs. Given a matrix $x \in \mathbb{R}^{n \times d}$, we let $x_c = \begin{bmatrix} x_1^T, & \dots, & x_n^T \end{bmatrix}^T \in \mathbb{R}^{nd}$ be the concatenation vector representing $x$. Given a vector $\theta \in \mathbb{R}^d$, we let $\theta_{c,n} = \begin{bmatrix} \theta^T, & \dots, & \theta^T \end{bmatrix}^T \in \mathbb{R}^{nd}$ be the concatenation of $n$ copies of $\theta$. Given a matrix $x \in \mathbb{R}^{n \times d}$, we let $\bar{x} \equiv \frac{1}{n} \sum_{i=1}^{n} x_i \in \mathbb{R}^d$ denote its mean along the sample dimension. For any matrix $B \in \mathbb{R}^{d \times d}$, we denote:

$$
diag(B, k) = \begin{bmatrix} B & & 0 \\ & \ddots & \\ 0 & & B \end{bmatrix} \in \mathbb{R}^{kd \times kd}
$$

and

$$rep(B, k) = \begin{bmatrix} B & \cdots & B \\ \vdots & \ddots & \vdots \\ B & \cdots & B \end{bmatrix} \in \mathbb{R}^{kd \times kd}$$

Finally, we define strategies for both attacker and authenticator, which we prove in what follows to be the optimal strategies for the game.

**Definition F.1.** *Let $\mathcal{G}^*$ denote an attacker defined by the following generative process: Given a leaked sample $Y \in \mathbb{R}^{m \times d}$, $\mathcal{G}^*$ generates an attack sample $X \in \mathbb{R}^{n \times d}$ as follows. It first samples $n$ vectors $W_1, \ldots, W_n \stackrel{iid}{\sim} \mathcal{N}(0, \Sigma)$ and then sets:*

$$X_i = W_i - \bar{W} + \bar{Y} \tag{F.1}$$

*Also, let $g^*_{X|Y}$ denote its associated conditional probability.*

**Definition F.2.** *For any $\alpha \in \mathbb{R}_+$, let $\mathcal{D}_\alpha$ denote an authenticator defined as:*

$$\mathcal{D}_\alpha(a, x) = I\left[\|\bar{x} - \bar{a}\|^2_{\Sigma^{-1}} < \alpha\right]$$

*Where $I$ is the indicator function*

### F.2 TECHNICAL LEMMAS

**Lemma F.3.** *Let $X_1, \ldots, X_k \stackrel{iid}{\sim} \mathcal{N}(\mu, \Sigma)$ Where $\mu \in \mathbb{R}^d, \Sigma \in \mathbb{R}^{d \times d}$. Then*

$$\bar{X} = \frac{1}{k} \sum_{j=1}^{k} X_j \sim \mathcal{N}(\mu, \frac{1}{k}\Sigma)$$

*Proof.* We begin by observing that $X_c \sim \mathcal{N}(\mu_{c,k}, diag(\Sigma, k))$. Let $B = \frac{1}{k}\begin{bmatrix} I_d & \cdots & I_d \end{bmatrix} \in \mathbb{R}^{d \times kd}$ and observe that $\bar{X} = BX_c$. Therefore, since this is an affine transformation of a Gaussian vector we have:

$$\bar{X} \sim \mathcal{N}(B\mu_{c,k}, Bdiag(\Sigma, k)B^T) = \mathcal{N}(\mu, \frac{1}{k}\Sigma)$$

As required. □

**Lemma F.4.** *Let $X \in \mathbb{R}^d$ be a Gaussian vector s.t $X \sim \mathcal{N}(\mu, \Sigma)$, and let $X_{c,n} \in \mathbb{R}^{nd}$ be the concatenation of $n$ copies of $X$. Then:*

$$X_{c,n} \sim \mathcal{N}(\mu_{c,n}, rep(\Sigma, n))$$

*Proof.* Let

$$B = \begin{bmatrix} I_d \\ \vdots \\ I_d \end{bmatrix} \in \mathbb{R}^{nd \times d}$$

and observe that $X_{c,n} = BX$. Therefore, since this is an affine transformation of a Gaussian vector we have:

$$X_{c,n} \sim \mathcal{N}(B\mu, B\Sigma B^T) = \mathcal{N}(\mu_{c,n}, rep(\Sigma, n))$$

As required. □

**Lemma F.5.** *Let $\theta \in \mathbb{R}^d, \Sigma \in \mathbb{R}^{d \times d}$ represent the mean and covariance of a Gaussian distribution. Let $X \in \mathbb{R}^{n \times d}$ be a random sample generated by the attacker defined in Def. F.1. Then:*

$$X_c \sim \mathcal{N}(\theta_{c,n}, diag(\Sigma, n) + rep((\frac{n-m}{mn})\Sigma, n)) \equiv \mathcal{N}(\theta_{c,n}, \Psi)$$

*Proof.* Observe that $W_c \sim \mathcal{N}(0, diag(\Sigma, n))$. Using Lemma F.3 we get $\bar{Y} \sim \mathcal{N}(\theta, \frac{1}{m}\Sigma)$ and observe that $\bar{W}_{c,n} = rep(\frac{1}{n}I_d, n)W_c$. Using Lemma F.4 we get $\bar{Y}_{c,n} \sim \mathcal{N}(\theta_{c,n}, \frac{1}{m}rep(\Sigma, n))$. We define the following block matrices

$$Z = \begin{bmatrix} W_c \\ \bar{Y}_{c,n} \end{bmatrix}, \quad B = \begin{bmatrix} I_{nd} - \frac{1}{n}rep(I_d, n), & I_{nd} \end{bmatrix}$$

and observe that:

$$Z \sim \mathcal{N}(\begin{bmatrix} 0_{nd} \\ \theta_{c,n} \end{bmatrix}, \begin{bmatrix} diag(\Sigma, n) & 0 \\ 0 & \frac{1}{m}rep(\Sigma, n) \end{bmatrix})$$

Note that $X_c = W_c - \bar{W}_{c,n} + \bar{Y}_{c,n} = BZ$ and therefore we get:

$$X_c \sim \mathcal{N}(B\begin{bmatrix} 0_{nd} \\ \theta_{c,n} \end{bmatrix}, B\begin{bmatrix} diag(\Sigma, n) & 0 \\ 0 & \frac{1}{m}rep(\Sigma, n) \end{bmatrix}B^T)$$

$$= \mathcal{N}\left(\theta_{c,n}, diag(\Sigma, n) + rep\left((\frac{n-m}{mn})\Sigma, n\right)\right)$$

As required. $\qquad\square$

**Lemma F.6.** *Let* $\Sigma = CC^T \in \mathbb{R}^{d\times d}$ *represent the covariance of a Gaussian distribution, and consider the following covariance matrix:*

$$\Psi = diag(\Sigma, n) + rep((\frac{n-m}{mn})\Sigma, n)$$

*Then its inverse is:*

$$\Psi^{-1} = diag(\Sigma^{-1}, n) - \frac{n-m}{n^2}rep(\Sigma^{-1}, n)$$

*and the determinant is:*

$$\det(\Psi) = \left(\frac{n}{m}\right)^d \det(\Sigma)^n$$

*Proof.* We begin by defining the following block matrices:

$$U = \begin{bmatrix} \Sigma \\ \vdots \\ \Sigma \end{bmatrix} \in \mathbb{R}^{nd\times d} \tag{F.2}$$

$$V = (\frac{n-m}{mn})\begin{bmatrix} I_d \\ \vdots \\ I_d \end{bmatrix} \in \mathbb{R}^{nd\times d} \tag{F.3}$$

and note that $rep((\frac{n-m}{mn})\Sigma, n) = UV^T$. Then:

$$
\begin{aligned}
\Psi^{-1} &= (diag(\Sigma, n) + rep((\frac{n-m}{mn})\Sigma, n))^{-1} \\
&= (diag(\Sigma, n) + UV^T)^{-1} \\
&\overset{(i)}{=} diag(\Sigma, n)^{-1} - diag(\Sigma, n)^{-1} U (I_d + V^T diag(\Sigma, n)^{-1} U)^{-1} V^T diag(\Sigma, n)^{-1} \\
&\overset{(ii)}{=} diag(\Sigma^{-1}, n) - diag(\Sigma^{-1}, n) U (I_d + V^T diag(\Sigma^{-1}, n) U)^{-1} V^T diag(\Sigma^{-1}, n) \\
&= diag(\Sigma^{-1}, n) - \begin{bmatrix} I_d \\ \vdots \\ I_d \end{bmatrix} (I_d + \frac{n-m}{nm} \begin{bmatrix} I_d & \cdots & I_d \end{bmatrix} \begin{bmatrix} I_d \\ \vdots \\ I_d \end{bmatrix})^{-1} \frac{n-m}{nm} \begin{bmatrix} \Sigma^{-1} & \cdots & \Sigma^{-1} \end{bmatrix} \\
&= diag(\Sigma^{-1}, n) - \begin{bmatrix} I_d \\ \vdots \\ I_d \end{bmatrix} \frac{m}{n} I_d \frac{n-m}{nm} \begin{bmatrix} \Sigma^{-1} & \cdots & \Sigma^{-1} \end{bmatrix} \\
&= diag(\Sigma^{-1}, n) - \frac{n-m}{n^2} \begin{bmatrix} I_d \\ \vdots \\ I_d \end{bmatrix} \begin{bmatrix} \Sigma^{-1} & \cdots & \Sigma^{-1} \end{bmatrix} \\
&= diag(\Sigma^{-1}, n) - \frac{n-m}{n^2} rep(\Sigma^{-1}, n)
\end{aligned}
$$

As required. Where in $(i)$ we used the Woodbury matrix identity, and in $(ii)$ we used the inverse of a diagonal block matrix.

Next, we turn to find the determinant of $\Psi$:

$$
\begin{aligned}
det(\Psi) &= det(diag(\Sigma, n) + rep((\frac{n-m}{mn})\Sigma, n)) \\
&= det(diag(\Sigma, n) + UV^T) \\
&\overset{(iii)}{=} det(diag(\Sigma, n)) det(I_d + V^T diag(\Sigma, n)^{-1} U) \\
&\overset{(iv)}{=} det(\Sigma)^n det(I_d + V^T diag(\Sigma^{-1}, n) U) \\
&= det(\Sigma)^n det(I_d + \frac{n-m}{nm} \begin{bmatrix} I_d & \cdots & I_d \end{bmatrix} diag(\Sigma^{-1}, n) \begin{bmatrix} \Sigma \\ \vdots \\ \Sigma \end{bmatrix}) \\
&= det(\Sigma)^n det(I_d + \frac{n-m}{m} I_d) \\
&= det(\Sigma)^n \left(\frac{n}{m}\right)^d
\end{aligned}
$$

Where in $(iii)$ we used the matrix determinant lemma, and in $(iv)$ we used the determinant of a diagonal block matrix. $\qquad \square$

**Lemma F.7.** *Let*
$$
h(x, \mu) = \exp\{-\frac{1}{2\sigma^2}(x-\mu)^T \Sigma^{-1}(x-\mu)\} I\left[x^T \Sigma^{-1} x < \alpha\right] \quad \forall(x, \mu) \in \mathbb{R}^d \times \mathbb{R}^d
$$
*and define the function:*
$$
\psi(\mu) = \int_{x \in \mathbb{R}^d} dx\, h(x, \mu)
$$
*Then $\psi(\mu)$ is log-concave over the space $\mathbb{R}^d$.*

*Proof.* We begin by noting that the function $(x-\mu)^T \Sigma^{-1}(x-\mu)$ is convex w.r.t both $x$ and $\mu$, hence its negative is concave, and thus, by definition the function:
$$
\exp\{-\frac{1}{2\sigma^2}(x-\mu)^T \Sigma^{-1}(x-\mu)\} \tag{F.4}
$$

is log-concave w.r.t $x, \mu$. We now show that $h(x, \mu)$ is log-concave w.r.t both $x$ and $\mu$. First, w.r.t $\mu$. Let $\beta \in [0, 1]$, $\mu_1, \mu_2 \in \mathbb{R}^d$, and observe that:

$$h(x, \beta\mu_1 + (1 - \beta)\mu_2)$$

$$= \exp\{-\frac{1}{2\sigma^2}(x - \beta\mu_1 - (1 - \beta)\mu_2)^T\Sigma^{-1}(x - \beta\mu_1 - (1 - \beta)\mu_2)\}I[x^T\Sigma^{-1}x < \alpha]$$

$$\overset{(i)}{\geqslant} \exp\{-\frac{\beta}{2\sigma^2}(x - \mu_1)^T\Sigma^{-1}(x - \mu_1)\}\exp\{-\frac{(1 - \beta)}{2\sigma^2}(x - \mu_2)^T\Sigma^{-1}(x - \mu_2)\}$$

$$I[x^T\Sigma^{-1}x < \alpha]$$

$$\overset{(ii)}{=} \exp\{-\frac{\beta}{2\sigma^2}(x - \mu_1)^T\Sigma^{-1}(x - \mu_1)\}\exp\{-\frac{(1 - \beta)}{2\sigma^2}(x - \mu_2)^T\Sigma^{-1}(x - \mu_2)\}$$

$$(I[x^T\Sigma^{-1}x < \alpha])^\beta(I[x^T\Sigma^{-1}x < \alpha])^{(1-\beta)}$$

$$= h(x, \mu_1)^\beta h(x, \mu_2)^{(1-\beta)}$$

Therefore $h(x, \mu)$ is log-concave w.r.t $\mu$. Where in $(i)$ we used the log-concavity of the function in Eq. F.4, and in $(ii)$ we used the fact that $I[x^T\Sigma^{-1}x < \alpha] \in \{0, 1\}$.

Now, w.r.t $x$. Let $\beta \in [0, 1]$, $x_1, x_2 \in \mathbb{R}^d$. Observe that for any convex function $q$ we have:

$$I[q(x_1) < \alpha]^\beta I[q(x_2) < \alpha]^{(1-\beta)} \overset{(i)}{=} I[q(x_1) < \alpha]I[q(x_2) < \alpha]$$

$$= I[q(x_1) < \alpha \wedge q(x_2) < \alpha]$$

$$\overset{(ii)}{\leqslant} I[\beta q(x_1) + (1 - \beta)q(x_2) < \alpha]$$

$$\overset{(iii)}{\leqslant} I[q(\beta x_1 + (1 - \beta)x_2) < \alpha]$$

Therefore $I[x^T\Sigma^{-1}x < \alpha]$ is log-concave. Where in $(i)$ we used the fact that $I[q(x) < \alpha] \in \{0, 1\}$, in $(ii)$ we used the fact that $q(x_1) < \alpha \wedge q(x_2) < \alpha \Rightarrow \beta q(x_1) + (1 - \beta)q(x_2) < \alpha$, and in $(iii)$ we used the convexity of $q$. Hence, observing $h(x, \mu)$ we have:

$$h(\beta x_1 + (1 - \beta)x_2, \mu)$$

$$= \exp\{-\frac{1}{2\sigma^2}(\beta x_1 + (1 - \beta)x_2 - \mu)^T\Sigma^{-1}(\beta x_1 + (1 - \beta)x_2 - \mu)\}$$

$$I[(\beta x_1 + (1 - \beta)x_2)^T\Sigma^{-1}(\beta x_1 + (1 - \beta)x_2) < \alpha]$$

$$\overset{(iv)}{\geqslant} \exp\{-\frac{\beta}{2\sigma^2}(x_1 - \mu)^T\Sigma^{-1}(x_1 - \mu)\}\exp\{-\frac{(1 - \beta)}{2\sigma^2}(x_2 - \mu)^T\Sigma^{-1}(x_2 - \mu)\}$$

$$I[(\beta x_1 + (1 - \beta)x_2)^T\Sigma^{-1}(\beta x_1 + (1 - \beta)x_2) < \alpha]$$

$$\overset{(v)}{\geqslant} \exp\{-\frac{\beta}{2\sigma^2}(x - \mu_1)^T\Sigma^{-1}(x - \mu_1)\}\exp\{-\frac{(1 - \beta)}{2\sigma^2}(x - \mu_2)^T\Sigma^{-1}(x - \mu_2)\}$$

$$(I[x_1^T\Sigma^{-1}x_1 < \alpha])^\beta(I[x_2^T\Sigma^{-1}x_2 < \alpha])^{(1-\beta)}$$

$$= h(x_1, \mu)^\beta h(x_2, \mu)^{(1-\beta)}$$

Therefore $h(x, \mu)$ is log-concave w.r.t $x$. Where in $(iv)$ we used the log-concavity of the function in Eq. F.4, and in $(v)$ we used the log-concavity of $I[x^T\Sigma^{-1}x < \alpha]$.

Finally, by using Prekopa-Leindler inequality (Prékopa, 1973) we have that $\psi(\mu)$ is log-concave, as required. □

### F.3 PROOF OF THEOREM 4.2

**Lemma F.8.** *Consider the attacker $\mathcal{G}^*$, defined in Def. F.1. The best response strategy for the authenticator against this attacker is:*

$$\mathcal{D}^*(a, x) = I\left[\|\bar{x} - \bar{a}\|^2_{\Sigma^{-1}} < \alpha^*\right]$$

*Where:*

$$\alpha^* = \frac{d(m + k)(n + k)}{k^2(n - m)}\log\frac{n(m + k)}{m(n + k)}$$

*Proof.* The best response authenticator satisfies:

$$\mathcal{D}^* \in \underset{\mathcal{D}\in\mathbb{D}}{\operatorname{argmax}} V(\mathcal{D}, \mathcal{G})$$

$$= \underset{\mathcal{D}\in\mathbb{D}}{\operatorname{argmax}} \frac{1}{2}\mathbb{E}_{\Theta\sim Q}\mathbb{E}_{A\sim f_\Theta^{(k)}}\mathbb{E}_{Y\sim f_\Theta^{(m)}}\big[\mathbb{E}_{X\sim f_\Theta^{(n)}}[\mathcal{D}(A, X)] + \mathbb{E}_{X\sim g_{X|Y}(\cdot|Y)}[1 - \mathcal{D}(A, X)]\big]$$

$$= \underset{\mathcal{D}\in\mathbb{D}}{\operatorname{argmax}} \mathbb{E}_{\Theta\sim Q}\mathbb{E}_{A\sim f_\Theta^{(k)}}\big[\mathbb{E}_{X\sim f_\Theta^{(n)}}[\mathcal{D}(A, X)] + \mathbb{E}_{X\sim g_{X|\Theta}(\cdot|\Theta)}[1 - \mathcal{D}(A, X)]\big]$$

$$\overset{F.5}{=} \underset{\mathcal{D}\in\mathbb{D}}{\operatorname{argmax}} \mathbb{E}_{\Theta\sim Q}\mathbb{E}_{A\sim\mathcal{N}(\Theta_{c,k}, diag(\Sigma, k))}$$

$$\big[\mathbb{E}_{X\sim\mathcal{N}(\Theta_{c,n}, diag(\Sigma, n))}[\mathcal{D}(A, X)] + \mathbb{E}_{X\sim\mathcal{N}(\theta_{c,n}, \Psi)}[1 - \mathcal{D}(A, X)]\big]$$

$$\overset{F.6}{=} \underset{\mathcal{D}\in\mathbb{D}}{\operatorname{argmax}} \mathbb{E}_{\Theta\sim Q} \int_{a\in\mathbb{R}^{kd}} da \int_{x\in\mathbb{R}^{nd}} dx \exp\{-\frac{1}{2}(a - \Theta_{c,k})^T diag(\Sigma, k)^{-1}(a - \Theta_{c,k})\}\big[$$

$$\exp\{-\frac{1}{2}(x - \Theta_{c,n})^T diag(\Sigma, n)^{-1}(x - \Theta_{c,n})\}\mathcal{D}(a, x)+$$

$$\sqrt{\frac{1}{(\frac{n}{m})^d}} \exp\{-\frac{1}{2}(x - \Theta_{c,n})^T\Psi^{-1}(x - \Theta_{c,n})\}[1 - \mathcal{D}(a, x)]\big]$$

$\mathcal{D}$ can be chosen independently for each pair $(a, x) \in \mathcal{X}^k \times \mathcal{X}^n$. Therefore, for any $(a, x) \in \mathcal{X}^k \times \mathcal{X}^n$ the decision rule for $\mathcal{D}(a, x) = 1$ is:

$$\sqrt{(\frac{n}{m})^d} \int_{\theta\in\mathbb{R}^d} d\theta Q(\theta) \exp\{-\frac{1}{2}[(x - \theta_{c,n})^T diag(\Sigma, n)^{-1}(x - \theta_{c,n})+$$

$$(a - \theta_{c,k})^T diag(\Sigma, k)^{-1}(a - \theta_{c,k})]\} >$$

$$\int_{\theta\in\mathbb{R}^d} d\theta Q(\theta) \exp\{-\frac{1}{2}[(x - \theta_{c,n})^T\Psi^{-1}(x - \theta_{c,n}) + (a - \theta_{c,k})^T diag(\Sigma, k)^{-1}(a - \theta_{c,k})]\}$$

Observing the LHS integral and using the improper uniform prior assumption we have:

$$\int_{\theta\in\mathbb{R}^d} d\theta Q(\theta) \exp\{-\frac{1}{2}[(x - \theta_{c,n})^T diag(\Sigma^{-1}, n)(x - \theta_{c,n})+$$

$$(a - \theta_{c,k})^T diag(\Sigma^{-1}, k)(a - \theta_{c,k})]\}$$

$$= \int_{\theta\in\mathbb{R}^d} d\theta \exp\{-\frac{1}{2}[\sum_{i=1}^n x_i^T\Sigma^{-1}x_i + \sum_{j=1}^k a_j^T\Sigma^{-1}a_j - 2\theta^T\Sigma^{-1}(n\bar{x} + k\bar{a}) + (n + k)\theta^T\Sigma^{-1}\theta]\}$$

$$= \exp\{-\frac{1}{2}[\sum_{i=1}^n x_i^T\Sigma^{-1}x_i + \sum_{j=1}^k a_j^T\Sigma^{-1}a_j - (n + k)(\frac{n\bar{x} + k\bar{a}}{n + k})^T\Sigma^{-1}(\frac{n\bar{x} + k\bar{a}}{n + k})]\}$$

$$\int_{\theta\in\mathbb{R}^d} d\theta \exp\{-\frac{n + k}{2}[(\theta - \frac{n\bar{x} + k\bar{a}}{n + k})^T\Sigma^{-1}(\theta - \frac{n\bar{x} + k\bar{a}}{n + k})]\}$$

$$= \exp\{-\frac{1}{2}[\sum_{i=1}^n x_i^T\Sigma^{-1}x_i + \sum_{j=1}^k a_j^T\Sigma^{-1}a_j - (n + k)(\frac{n\bar{x} + k\bar{a}}{n + k})^T\Sigma^{-1}(\frac{n\bar{x} + k\bar{a}}{n + k})]\}$$

$$\sqrt{(\frac{2\pi}{n + k})^d \det(\Sigma)}$$

Observing the RHS and using the improper uniform prior assumption we have:

$$\int_{\theta\in\mathbb{R}^d} d\theta Q(\theta) \exp\{-\frac{1}{2}[(x - \theta_{c,n})^T\Psi^{-1}(x - \theta_{c,n}) + (a - \theta_{c,k})^T diag(\Sigma^{-1}, k)(a - \theta_{c,k})]\}$$

$$= \int_{\theta\in\mathbb{R}^d} d\theta \exp\{-\frac{1}{2}[(x - \theta_{c,n})^T\Psi^{-1}(x - \theta_{c,n}) + (a - \theta_{c,k})^T diag(\Sigma^{-1}, k)(a - \theta_{c,k})]\}$$

$$\overset{F.6}{=} \int_{\theta\in\mathbb{R}^d} d\theta \exp\{-\frac{1}{2}[(x - \theta_{c,n})^T(diag(\Sigma^{-1}, n) - \frac{n - m}{n^2}rep(\Sigma^{-1}, n))(x - \theta_{c,n})+$$

$$(a - \theta_{c,k})^T diag(\Sigma^{-1}, k)(a - \theta_{c,k})]\}$$

$$= \int_{\theta \in \mathbb{R}^d} d\theta \exp\{-\frac{1}{2}[\sum_{i=1}^{n}(x_i - \theta)^T\Sigma^{-1}(x_i - \theta) + \sum_{j=1}^{k}(a_j - \theta)^T\Sigma^{-1}(a_j - \theta) -$$

$$\frac{n-m}{n^2}\sum_{i=1}^{n}\sum_{j=1}^{n}(x_i - \theta)^T\Sigma^{-1}(x_j - \theta)]\}$$

$$= \int_{\theta \in \mathbb{R}^d} d\theta \exp\{-\frac{1}{2}[\sum_{i=1}^{n}(x_i - \theta)^T\Sigma^{-1}(x_i - \theta) + \sum_{j=1}^{k}(a_j - \theta)^T\Sigma^{-1}(a_j - \theta) -$$

$$(n-m)(\bar{x} - \theta)^T\Sigma^{-1}(\bar{x} - \theta)]\}$$

$$= \int_{\theta \in \mathbb{R}^d} d\theta \exp\{-\frac{1}{2}[\sum_{i=1}^{n}x_i^T\Sigma^{-1}x_i + \sum_{j=1}^{k}a_j^T\Sigma^{-1}a_j - (n-m)\bar{x}^T\Sigma^{-1}\bar{x} +$$

$$(m+k)\theta^T\Sigma^{-1}\theta - 2\theta^T\Sigma^{-1}(m\bar{x} + k\bar{a})]\}$$

$$= \exp\{-\frac{1}{2}[\sum_{i=1}^{n}x_i^T\Sigma^{-1}x_i + \sum_{j=1}^{k}a_j^T\Sigma^{-1}a_j - (n-m)\bar{x}^T\Sigma^{-1}\bar{x} -$$

$$(m+k)(\frac{m\bar{x} + k\bar{a}}{m+k})^T\Sigma^{-1}(\frac{m\bar{x} + k\bar{a}}{m+k})]\}$$

$$\int_{\theta \in \mathbb{R}^d} d\theta \exp\{-\frac{m+k}{2}[(\theta - \frac{m\bar{x} + k\bar{a}}{m+k})^T\Sigma^{-1}(\theta - \frac{m\bar{x} + k\bar{a}}{m+k})]\}$$

$$= \sqrt{(\frac{2\pi}{m+k})^d \det(\Sigma)} \exp\{-\frac{1}{2}[\sum_{i=1}^{n}x_i^T\Sigma^{-1}x_i + \sum_{j=1}^{k}a_j^T\Sigma^{-1}a_j - (n-m)\bar{x}^T\Sigma^{-1}\bar{x} -$$

$$(m+k)(\frac{m\bar{x} + k\bar{a}}{m+k})^T\Sigma^{-1}(\frac{m\bar{x} + k\bar{a}}{m+k})]\}$$

Therefore the decision rule is

$$\sqrt{(\frac{n}{m})^d} \exp\{\frac{1}{2}[(n+k)(\frac{n\bar{x} + k\bar{a}}{n+k})^T\Sigma^{-1}(\frac{n\bar{x} + k\bar{a}}{n+k})]\}\sqrt{(\frac{2\pi}{n+k})^d} >$$

$$\exp\{\frac{1}{2}[(n-m)\bar{x}^T\Sigma^{-1}\bar{x} + (m+k)(\frac{m\bar{x} + k\bar{a}}{m+k})^T\Sigma^{-1}(\frac{m\bar{x} + k\bar{a}}{m+k})]\}\sqrt{(\frac{2\pi}{m+k})^d}$$

$$\Leftrightarrow \sqrt{(\frac{n(m+k)}{m(n+k)})^d} \exp\{\frac{1}{2}[(n+k)(\frac{n\bar{x} + k\bar{a}}{n+k})^T\Sigma^{-1}(\frac{n\bar{x} + k\bar{a}}{n+k})]\} >$$

$$\exp\{\frac{1}{2}[(n-m)\bar{x}^T\Sigma^{-1}\bar{x} + (m+k)(\frac{m\bar{x} + k\bar{a}}{m+k})^T\Sigma^{-1}(\frac{m\bar{x} + k\bar{a}}{m+k})]\}$$

$$\Leftrightarrow d\log\frac{n(m+k)}{m(n+k)} > (n-m)\bar{x}^T\Sigma^{-1}\bar{x} + (m+k)(\frac{m\bar{x} + k\bar{a}}{m+k})^T\Sigma^{-1}(\frac{m\bar{x} + k\bar{a}}{m+k}) -$$

$$(n+k)(\frac{n\bar{x} + k\bar{a}}{n+k})^T\Sigma^{-1}(\frac{n\bar{x} + k\bar{a}}{n+k})$$

$$\Leftrightarrow d(m+k)(n+k)\log\frac{n(m+k)}{m(n+k)} >$$

$$k^2(n-m)\bar{x}^T\Sigma^{-1}\bar{x} - 2k^2(n-m)\bar{x}^T\Sigma^{-1}\bar{a} + k^2(n-m)\bar{a}^T\Sigma^{-1}\bar{a}$$

$$\Leftrightarrow \frac{d(m+k)(n+k)}{k^2(n-m)}\log\frac{n(m+k)}{m(n+k)} > (\bar{x} - \bar{a})^T\Sigma^{-1}(\bar{x} - \bar{a})$$

As required. $\square$

**Lemma F.9.** *Consider the authenticator $\mathcal{D}_\alpha$, as defined in Def. F.2. Then any attacker $\mathcal{G}$, represented by a conditional probability $g_{X|Y}$, that satisfies the condition $\bar{x} = \bar{y}$ for any leaked sample $y \in \mathbb{R}^{m \times d}$ and attacker generated sample $x \in \{\mathbb{R}^{n \times d} : g_{X|Y}(x|y) > 0\}$, satisfies:*

$$\mathcal{G} \in \operatorname*{argmin}_{\mathcal{G}' \in \mathbb{G}} V(\mathcal{D}_\alpha, \mathcal{G}') \quad \forall \alpha \in \mathbb{R}_+$$

*Proof.* The best response attacker satisfies:

$$g'_{X|Y} \in \operatorname*{argmin}_{g_{X|Y}} \frac{1}{2} \mathbb{E}_{\Theta \sim Q} \mathbb{E}_{A \sim f_\Theta^{(k)}} \mathbb{E}_{Y \sim f_\Theta^{(m)}} [\mathbb{E}_{X \sim f_\Theta^{(n)}} [\mathcal{D}_\alpha(A, X)] + \mathbb{E}_{X \sim g_{X|Y}(\cdot|Y)} [1 - \mathcal{D}_\alpha(A, X)]]$$

$$= \operatorname*{argmin}_{g_{X|Y}} \mathbb{E}_{\Theta \sim Q} \mathbb{E}_{A \sim f_\Theta^{(k)}} \mathbb{E}_{Y \sim f_\Theta^{(m)}} \mathbb{E}_{X \sim g_{X|Y}(\cdot|Y)} [1 - \mathcal{D}_\alpha(A, X)]$$

$$= \operatorname*{argmax}_{g_{X|Y}} \mathbb{E}_{\Theta \sim Q} \mathbb{E}_{A \sim f_\Theta^{(k)}} \mathbb{E}_{Y \sim f_\Theta^{(m)}} \mathbb{E}_{X \sim g_{X|Y}(\cdot|Y)} [\mathcal{D}_\alpha(A, X)]$$

$$= \operatorname*{argmax}_{g_{X|Y}} \mathbb{E}_{\Theta \sim Q} \mathbb{E}_{A \overset{\text{iid}}{\sim} \mathcal{N}(\Theta, \Sigma)} \mathbb{E}_{Y \overset{\text{iid}}{\sim} \mathcal{N}(\Theta, \Sigma)} \mathbb{E}_{X \sim g_{X|Y}(\cdot|Y)} \left[ I \left[ \left\| \bar{X} - \bar{A} \right\|_{\Sigma^{-1}}^2 < \alpha \right] \right]$$

$$\overset{\text{Lemma F.3}}{=} \operatorname*{argmax}_{g_{X|Y}} \mathbb{E}_{\Theta \sim Q} \mathbb{E}_{\bar{A} \sim \mathcal{N}(\Theta, \frac{1}{k}\Sigma)} \mathbb{E}_{Y \overset{\text{iid}}{\sim} \mathcal{N}(\Theta, \Sigma)} \mathbb{E}_{X \sim g_{X|Y}(\cdot|Y)} \left[ I \left[ \left\| \bar{X} - \bar{A} \right\|_{\Sigma^{-1}}^2 < \alpha \right] \right]$$

$$= \operatorname*{argmax}_{g_{X|Y}} \int_{y \in \mathbb{R}^{m \times d}} dy \int_{x \in \mathbb{R}^{n \times d}} dx g_{X|Y}(x|y) \int_{\bar{a} \in \mathbb{R}^d} d\bar{a} I \left[ \left\| \bar{x} - \bar{a} \right\|_{\Sigma^{-1}}^2 < \alpha \right] \int_{\theta \in \mathbb{R}^d} d\theta Q(\theta)$$

$$\exp\{-\frac{k}{2} (\bar{a} - \theta)^T \Sigma^{-1} (\bar{a} - \theta)\} \exp\{-\frac{1}{2} \sum_{j=1}^m (y_j - \theta)^T \Sigma^{-1} (y_j - \theta)\}$$

Note that $g_{X|Y}(x|y)$ can be chosen independently for each $y \in \mathbb{R}^{m \times d}$. Thus, we can optimize it independently for each $y \in \mathbb{R}^{m \times d}$ and we have:

$$g'_{X|Y}(\cdot|y) \in \operatorname*{argmax}_{g_{X|Y}(\cdot|y)} \int_{x \in \mathbb{R}^{n \times d}} dx g_{X|Y}(x|y) \int_{\bar{a} \in \mathbb{R}^d} d\bar{a} I \left[ \left\| \bar{x} - \bar{a} \right\|_{\Sigma^{-1}}^2 < \alpha \right] \int_{\theta \in \mathbb{R}^d} d\theta Q(\theta)$$

$$\exp\{-\frac{k}{2} (\bar{a} - \theta)^T \Sigma^{-1} (\bar{a} - \theta)\} \exp\{-\frac{1}{2} \sum_{j=1}^m (y_j - \theta)^T \Sigma^{-1} (y_j - \theta)\}$$

Note that for any PDF $f$ over $\mathbb{R}^{n \times d}$ and a function $\varphi : \mathbb{R}^{n \times d} \rightarrow \mathbb{R}$, it holds that $\int_{x \in \mathbb{R}^{n \times d}} dx f(x) \varphi(x) \leqslant sup_x \varphi(x)$. Therefore, there exists a deterministic distribution $g'_{X|Y}(x|y) = \delta(x - x')$ that achieves the maximum. Thus, it's sufficient to find a vector $x_{\mathcal{G}}$ that achieves the maximum:

$$x_{\mathcal{G}} \in \operatorname*{argmax}_x \int_{x' \in \mathbb{R}^{n \times d}} dx' \delta(x' - x) \int_{\bar{a} \in \mathbb{R}^d} d\bar{a} I \left[ \left\| \bar{x}' - \bar{a} \right\|_{\Sigma^{-1}}^2 < \alpha \right] \int_{\theta \in \mathbb{R}^d} d\theta Q(\theta)$$

$$\exp\{-\frac{k}{2} (\bar{a} - \theta)^T \Sigma^{-1} (\bar{a} - \theta)\} \exp\{-\frac{1}{2} \sum_{j=1}^m (y_j - \theta)^T \Sigma^{-1} (y_j - \theta)\}$$

$$= \operatorname*{argmax}_x \int_{\bar{a} \in \mathbb{R}^d} d\bar{a} I \left[ \left\| \bar{x} - \bar{a} \right\|_{\Sigma^{-1}}^2 < \alpha \right] \int_{\theta \in \mathbb{R}^d} d\theta Q(\theta)$$

$$\exp\{-\frac{k}{2} (\bar{a} - \theta)^T \Sigma^{-1} (\bar{a} - \theta)\} \exp\{-\frac{1}{2} \sum_{j=1}^m (y_j - \theta)^T \Sigma^{-1} (y_j - \theta)\}$$

$$\overset{(*)}{=} \operatorname*{argmax}_x \int_{\bar{a} \in \mathbb{R}^d} d\bar{a} I \left[ \left\| \bar{x} - \bar{a} \right\|_{\Sigma^{-1}}^2 < \alpha \right] \int_{\theta \in \mathbb{R}^d} d\theta$$

$$\exp\{-\frac{1}{2}[k(\bar{a} - \theta)^T \Sigma^{-1} (\bar{a} - \theta) + \sum_{j=1}^m (y_j - \theta)^T \Sigma^{-1} (y_j - \theta)]\}$$

$$= \operatorname*{argmax}_x \int_{\bar{a} \in \mathbb{R}^d} d\bar{a} \exp\{-\frac{1}{2}[k\bar{a}^T \Sigma^{-1} \bar{a}]\} I \left[ \left\| \bar{x} - \bar{a} \right\|_{\Sigma^{-1}}^2 < \alpha \right] \int_{\theta \in \mathbb{R}^d} d\theta$$

$$\exp\{-\frac{1}{2}[(m+k)\theta^T \Sigma^{-1} \theta - 2\theta^T \Sigma^{-1} (m\bar{y} + k\bar{a})]\}$$

$$= \operatorname*{argmax}_x \int_{\bar{a} \in \mathbb{R}^d} d\bar{a} \exp\{-\frac{1}{2}[k\bar{a}^T \Sigma^{-1} \bar{a} - \frac{1}{m+k} (m\bar{y} + k\bar{a})^T \Sigma^{-1} (m\bar{y} + k\bar{a})]\}$$

$$I \left[ \left\| \bar{x} - \bar{a} \right\|_{\Sigma^{-1}}^2 < \alpha \right] \int_{\theta \in \mathbb{R}^d} d\theta \exp\{-\frac{(m+k)}{2} (\theta - \frac{m\bar{y} + k\bar{a}}{m+k})^T \Sigma^{-1} (\theta - \frac{m\bar{y} + k\bar{a}}{m+k})\}$$

$$= \underset{x}{\operatorname{argmax}} \int_{\bar{a} \in \mathbb{R}^d} d\bar{a} \exp\{-\frac{1}{2}[k\bar{a}^T \Sigma^{-1} \bar{a} - \frac{1}{m+k}(m\bar{y} + k\bar{a})^T \Sigma^{-1}(m\bar{y} + k\bar{a})]\}$$
$$I\left[\|\bar{x} - \bar{a}\|_{\Sigma^{-1}}^2 < \alpha\right]$$
$$= \underset{x}{\operatorname{argmax}} \int_{\bar{a} \in \mathbb{R}^d} d\bar{a} \exp\{-\frac{1}{2}[\frac{mk}{m+k}\bar{a}^T \Sigma^{-1} \bar{a} - \frac{2mk}{m+k}\bar{y}^T \Sigma^{-1} \bar{a} + \frac{mk}{m+k}\bar{y}^T \Sigma^{-1} \bar{y}]\}$$
$$I\left[\|\bar{x} - \bar{a}\|_{\Sigma^{-1}}^2 < \alpha\right]$$
$$= \underset{x}{\operatorname{argmax}} \int_{\bar{a} \in \mathbb{R}^d} d\bar{a} \exp\{-\frac{mk}{2(m+k)}[(\bar{a} - \bar{y})^T \Sigma^{-1}(\bar{a} - \bar{y})]\} I\left[\|\bar{x} - \bar{a}\|_{\Sigma^{-1}}^2 < \alpha\right]$$

Where in $(*)$ we used the fact that $Q(\theta)$ is the improper uniform prior. Note that the expression depends only on the mean $\bar{x}$. Therefore, it's sufficient to find a mean vector $\bar{x}$ that maximizes the expression. We substitute the integration variable to $\varphi = \bar{a} - \bar{x}$ and obtain:

$$\bar{x}_{\mathcal{G}} \in \underset{\bar{x}}{\operatorname{argmax}} \int_{\{\varphi \in \mathbb{R}^d : \varphi^T \Sigma^{-1} \varphi < \alpha\}} d\varphi \exp\{-\frac{mk}{2(m+k)}[(\varphi + \bar{x} - \bar{y})^T \Sigma^{-1}(\varphi + \bar{x} - \bar{y})]\}$$
$$\equiv \underset{\bar{x}}{\operatorname{argmax}} \psi(\bar{y} - \bar{x}) \equiv \underset{\bar{x}}{\operatorname{argmax}} \psi(\mu)$$

Where $\psi$ is defined as in Lemma F.7 (with $\sigma = \frac{m+k}{mk}$), from which it follows that $\psi(\mu)$ is log-concave, and therefore has at most one local extremum which can only be a maximum. Therefore, it is sufficient to show that $\mu = 0$ (i.e., $\bar{x} = \bar{y}$) is a local extremum by equating the gradient at the point to 0.

$$\frac{\partial}{\partial \mu} \psi(\mu) = \frac{\partial}{\partial \mu} \int_{\{\varphi \in \mathbb{R}^d : \varphi^T \Sigma^{-1} \varphi < \alpha\}} d\varphi \exp\{-\frac{mk}{2(m+k)}[(\varphi - \mu)^T \Sigma^{-1}(\varphi - \mu)]\}$$
$$= -\frac{mk}{2(m+k)} \int_{\{\varphi \in \mathbb{R}^d : \varphi^T \Sigma^{-1} \varphi < \alpha\}} d\varphi \exp\{-\frac{mk}{2(m+k)}[(\varphi - \mu)^T \Sigma^{-1}(\varphi - \mu)]\}$$
$$\frac{\partial}{\partial \mu}(\varphi - \mu)^T \Sigma^{-1}(\varphi - \mu)$$
$$= -\frac{mk}{(m+k)} \int_{\{\varphi \in \mathbb{R}^d : \varphi^T \Sigma^{-1} \varphi < \alpha\}} d\varphi \exp\{-\frac{mk}{2(m+k)}[(\varphi - \mu)^T \Sigma^{-1}(\varphi - \mu)]\} \Sigma^{-1}(\mu - \varphi)$$

Therefore:

$$\frac{\partial}{\partial \mu} \psi(\mu)|_{\mu=0} = \frac{mk}{(m+k)} \int_{\{\varphi \in \mathbb{R}^d : \varphi^T \Sigma^{-1} \varphi < \alpha\}} d\varphi \exp\{-\frac{mk}{2(m+k)}[\varphi^T \Sigma^{-1} \varphi]\} \Sigma^{-1} \varphi$$

Note that since the domain of integration is symmetric about the origin with respect to negation and the integrand is odd with respect to the integration variable, the integral is equal to zero. I.e., $\frac{\partial}{\partial \mu} \psi(\mu)|_{\mu=0} = 0$. Therefore, $\bar{x} = \bar{y}$ ($\mu = 0$) achieves the global maximum, and any attacker that satisfies the condition: $\bar{x} = \bar{y}$ for any leaked sample $y \in \mathbb{R}^{m \times d}$ and attacker generated sample $x \in \{\mathbb{R}^{n \times d} : g_{X|Y}(x|y) > 0\}$ satisfies:

$$\mathcal{G} \in \underset{\mathcal{G}' \in \mathbb{G}}{\operatorname{argmin}} V(\mathcal{D}_\alpha, \mathcal{G}') \quad \forall \alpha \in \mathbb{R}_+$$

As required. □

**Corollary F.10.** *Consider an authenticator $\mathcal{D}_\alpha$, as defined in Def. F.2. Then the attacker $\mathcal{G}^*$, defined in Def. F.1, is a best response. i.e.:*

$$\mathcal{G}^* \in \underset{\mathcal{G}' \in \mathbb{G}}{\operatorname{argmin}} V(\mathcal{D}_\alpha, \mathcal{G}') \quad \forall \alpha \in \mathbb{R}_+$$

*Proof.* Directly from Lemma F.9 □

**Theorem F.11.** *The game value is:*

$$\max_{\mathcal{D}} \min_{\mathcal{G}} V(\mathcal{D}, \mathcal{G}) = \min_{\mathcal{G}} \max_{\mathcal{D}} V(\mathcal{D}, \mathcal{G}) = V(\mathcal{D}^*, \mathcal{G}^*) =$$

$$\frac{1}{2} + \frac{1}{2\Gamma(\frac{d}{2})}[\gamma(\frac{d}{2}, \frac{dn(m+k)}{2k(n-m)} \log \frac{n(m+k)}{m(n+k)}) - \gamma(\frac{d}{2}, \frac{dm(n+k)}{2k(n-m)} \log \frac{n(m+k)}{m(n+k)})]$$

*Where $\gamma$ is the lower incomplete gamma function.*

*Proof.* From the max-min inequality we have:

$$\max_{\mathcal{D}} \min_{\mathcal{G}} V(\mathcal{D}, \mathcal{G}) \leqslant \min_{\mathcal{G}} \max_{\mathcal{D}} V(\mathcal{D}, \mathcal{G})$$

On the other hand, using Lemma F.8 and Corollary F.10 we have:

$$\max_{\mathcal{D}} \min_{\mathcal{G}} V(\mathcal{D}, \mathcal{G}) \geqslant \min_{\mathcal{G}} V(\mathcal{D}^*, \mathcal{G}) \stackrel{F.10}{=} V(\mathcal{D}^*, \mathcal{G}^*) \stackrel{F.8}{=} \max_{\mathcal{D}} V(\mathcal{D}, \mathcal{G}^*) \geqslant \min_{\mathcal{G}} \max_{\mathcal{D}} V(\mathcal{D}, \mathcal{G})$$

Therefore:

$$\max_{\mathcal{D}} \min_{\mathcal{G}} V(\mathcal{D}, \mathcal{G}) = \min_{\mathcal{G}} \max_{\mathcal{D}} V(\mathcal{D}, \mathcal{G}) = V(\mathcal{D}^*, \mathcal{G}^*)$$

The game value is given by:

$$
\begin{aligned}
V(\mathcal{D}^*, \mathcal{G}^*) =& \mathbb{E}_{\Theta \sim Q} V(\Theta, \mathcal{D}^*, \mathcal{G}^*) \\
=& \frac{1}{2} \mathbb{E}_{\Theta \sim Q} \mathbb{E}_{A \sim f_\Theta^{(k)}} \mathbb{E}_{Y \sim f_\Theta^{(m)}} \left[ \mathbb{E}_{X \sim f_\Theta^{(n)}} [\mathcal{D}^*(A, X)] + \mathbb{E}_{X \sim g^*_{X|Y}(\cdot|Y)}[1 - \mathcal{D}^*(A, X)] \right] \\
=& \frac{1}{2} \mathbb{E}_{\Theta \sim Q} \mathbb{E}_{A \sim f_\Theta^{(k)}} \mathbb{E}_{Y \sim f_\Theta^{(m)}} \\
& \left[ \mathbb{E}_{X \sim f_\Theta^{(n)}} \left[ I[\|\bar{X} - \bar{A}\|^2_{\Sigma^{-1}} < \alpha^*] \right] + \mathbb{E}_{X \sim g^*_{X|Y}(\cdot|Y)} \left[ 1 - I[\|\bar{X} - \bar{A}\|^2_{\Sigma^{-1}} < \alpha^*] \right] \right] \\
=& \frac{1}{2} + \frac{1}{2} \mathbb{E}_{\Theta \sim Q} \mathbb{E}_{A \sim f_\Theta^{(k)}} \mathbb{E}_{X \sim f_\Theta^{(n)}} \left[ I[\|\bar{x} - \bar{A}\|^2_{\Sigma^{-1}} < \alpha^*] \right] - \\
& \frac{1}{2} \mathbb{E}_{\Theta \sim Q} \mathbb{E}_{A \sim f_\Theta^{(k)}} \mathbb{E}_{X \sim g^*_{X|\Theta}(\cdot|\Theta)} \left[ I[\|\bar{x} - \bar{A}\|^2_{\Sigma^{-1}} < \alpha^*] \right]
\end{aligned}
$$

Observing the first term we have:

$$
\begin{aligned}
& \frac{1}{2} \mathbb{E}_{\Theta \sim Q} \mathbb{E}_{A \sim f_\Theta^{(k)}} \mathbb{E}_{X \sim f_\Theta^{(n)}} \left[ I[\|\bar{X} - \bar{A}\|^2_{\Sigma^{-1}} < \alpha^*] \right] \\
=& \frac{1}{2} \mathbb{E}_{\Theta \sim Q} \mathbb{E}_{A \sim f_\Theta^{(k)}} \mathbb{E}_{X \sim f_\Theta^{(n)}} \left[ I[(\bar{X} - \bar{A})^T \Sigma^{-1} (\bar{X} - \bar{A}) < \alpha^*] \right] \\
=& \frac{1}{2} \mathbb{E}_{\Theta \sim Q} \mathbb{E}_{A \sim f_\Theta^{(k)}} \mathbb{E}_{X \sim f_\Theta^{(n)}} \left[ I[(\bar{X} - \bar{A})^T (CC^T)^{-1} (\bar{X} - \bar{A}) < \alpha^*] \right] \\
=& \frac{1}{2} \mathbb{E}_{\Theta \sim Q} \mathbb{E}_{A \sim f_\Theta^{(k)}} \mathbb{E}_{X \sim f_\Theta^{(n)}} \left[ I[(\bar{X} - \bar{A})^T C^{-T} C^{-1} (\bar{X} - \bar{A}) < \alpha^*] \right] \\
=& \frac{1}{2} \mathbb{E}_{\Theta \sim Q} \mathbb{E}_{A \sim f_\Theta^{(k)}} \mathbb{E}_{X \sim f_\Theta^{(n)}} \left[ I[(C^{-1}(\bar{X} - \bar{A}))^T (C^{-1}(\bar{X} - \bar{A})) < \alpha^*] \right] \\
\equiv& \frac{1}{2} \mathbb{E}_{\Theta \sim Q} \mathbb{E}_{A \sim f_\Theta^{(k)}} \mathbb{E}_{X \sim f_\Theta^{(n)}} \left[ I[Z^T Z < \alpha^*] \right] \\
=& (*)
\end{aligned}
$$

Observe that

$$
\begin{aligned}
Z = C^{-1}(\bar{X} - \bar{A}) &= C^{-1}[(\bar{X} - \Theta) - (\bar{A} - \Theta)] \\
&= C^{-1}[I_d, -I_d] \begin{bmatrix} \bar{X} - \Theta \\ \bar{A} - \Theta \end{bmatrix} = [C^{-1}, -C^{-1}] \begin{bmatrix} \bar{X} - \Theta \\ \bar{A} - \Theta \end{bmatrix}
\end{aligned}
$$

Note that:

$$
\begin{bmatrix} \bar{X} - \Theta \\ \bar{A} - \Theta \end{bmatrix} \sim \mathcal{N}(0_{2d}, \begin{bmatrix} \frac{1}{n}\Sigma & 0_{d \times d} \\ 0_{d \times d} & \frac{1}{k}\Sigma \end{bmatrix})
$$

Therefore:

$$
\begin{aligned}
Z &\sim \mathcal{N}(0_d, [C^{-1}, -C^{-1}] \begin{bmatrix} \frac{1}{n}\Sigma & 0_{d \times d} \\ 0_{d \times d} & \frac{1}{k}\Sigma \end{bmatrix} \begin{bmatrix} C^{-T} \\ -C^{-T} \end{bmatrix}) \\
&= \mathcal{N}(0_d, (\frac{1}{n} + \frac{1}{k}) C^{-1} \Sigma C^{-T}) \\
&= \mathcal{N}(0_d, \frac{n+k}{nk} C^{-1} CC^T C^{-T}) \\
&= \mathcal{N}(0_d, \frac{n+k}{nk} I_d)
\end{aligned}
$$

We denote $\tilde{Z} = \sqrt{\frac{nk}{n+k}} Z \sim \mathcal{N}(0_d, I_d)$, and thus $\tilde{Z}_1, \ldots, \tilde{Z}_d$ are independent standard normal random variables and $\tilde{Z}^T \tilde{Z} \sim \chi^2(d)$. Therefore, $Z^T Z = \frac{n+k}{nk} \tilde{Z}^T \tilde{Z} \sim \Gamma(k = \frac{d}{2}, \theta = 2\frac{n+k}{nk})$ and we have:

$$
\begin{aligned}
(*) &= \frac{1}{2} \mathbb{E}_{\Theta \sim Q} \mathbb{E}_{A \sim f_\Theta^{(k)}} \mathbb{E}_{X \sim f_\Theta^{(n)}} \left[ I[Z^T Z < \alpha^*] \right] \\
&= \frac{1}{2} \mathbb{E}_{\Theta \sim Q} \mathbb{E}_{Z^T Z \sim \Gamma(k=\frac{d}{2}, \theta=2\frac{n+k}{nk})} \left[ I[Z^T Z < \alpha^*] \right] \\
&\overset{(i)}{=} \frac{1}{2} \mathbb{E}_{\Theta \sim Q} \frac{1}{\Gamma(\frac{d}{2})} \gamma(\frac{d}{2}, \frac{nk\alpha^*}{2(n+k)}) \\
&= \frac{1}{2} \frac{1}{\Gamma(\frac{d}{2})} \gamma(\frac{d}{2}, \frac{nk\alpha^*}{2(n+k)})
\end{aligned}
$$

Where in $(i)$ We used the CDF of the Gamma distribution in which $\gamma$ is the lower incomplete gamma function.

Similarly, observing the second term we have:

$$
\begin{aligned}
&\frac{1}{2} \mathbb{E}_{\Theta \sim Q} \mathbb{E}_{A \sim f_\Theta^{(k)}} \mathbb{E}_{X \sim g_{X|\Theta}^*(\cdot|\Theta)} \left[ I[\|\bar{X} - \bar{A}\|_{\Sigma^{-1}}^2 < \alpha^*] \right] \\
\equiv &\frac{1}{2} \mathbb{E}_{\Theta \sim Q} \mathbb{E}_{A \sim f_\Theta^{(k)}} \mathbb{E}_{X \sim g_{X|\Theta}^*(\cdot|\Theta)} \left[ I[V^T V < \alpha^*] \right] \\
= &(**)
\end{aligned}
$$

Where:

$$
\begin{aligned}
V &= C^{-1}(\bar{X} - \bar{A}) = C^{-1}[(\bar{X} - \Theta) - (\bar{A} - \Theta)] \\
&= C^{-1}[I_d, -I_d] \begin{bmatrix} \bar{X} - \Theta \\ \bar{A} - \Theta \end{bmatrix} = [C^{-1}, -C^{-1}] \begin{bmatrix} \bar{X} - \Theta \\ \bar{A} - \Theta \end{bmatrix}
\end{aligned}
$$

Using the definition of $\mathcal{G}^*$ (Definition F.1) and Lemma F.3 we have:

$$
\begin{bmatrix} \bar{X} - \Theta \\ \bar{A} - \Theta \end{bmatrix} \sim \mathcal{N}(0_{2d}, \begin{bmatrix} \frac{1}{m}\Sigma & 0_{d \times d} \\ 0_{d \times d} & \frac{1}{k}\Sigma \end{bmatrix})
$$

Therefore:

$$
V \sim \mathcal{N}(0_d, [C^{-1}, -C^{-1}] \begin{bmatrix} \frac{1}{m}\Sigma & 0_{d \times d} \\ 0_{d \times d} & \frac{1}{k}\Sigma \end{bmatrix} \begin{bmatrix} C^{-T} \\ -C^{-T} \end{bmatrix}) = \mathcal{N}(0_d, \frac{m+k}{mk} I_d)
$$

And similarly to the first term, we get:

$$
V^T V \sim \Gamma(k = \frac{d}{2}, \theta = 2\frac{m+k}{mk})
$$

And thus:

$$
(**) = \frac{1}{2} \frac{1}{\Gamma(\frac{d}{2})} \gamma(\frac{d}{2}, \frac{mk\alpha^*}{2(m+k)})
$$

Therefore, the game value is given by:

$$
\begin{aligned}
V(\mathcal{D}^*, \mathcal{G}^*) &= \frac{1}{2} + \frac{1}{2} \frac{1}{\Gamma(\frac{d}{2})} [\gamma(\frac{d}{2}, \frac{nk\alpha^*}{2(n+k)}) - \gamma(\frac{d}{2}, \frac{mk\alpha^*}{2(m+k)})] \\
&= \frac{1}{2} + \frac{1}{2} \frac{1}{\Gamma(\frac{d}{2})} [\gamma(\frac{d}{2}, \frac{dn(m+k)}{2k(n-m)} \log \frac{n(m+k)}{m(n+k)}) - \gamma(\frac{d}{2}, \frac{dm(n+k)}{2k(n-m)} \log \frac{n(m+k)}{m(n+k)})]
\end{aligned}
$$

As required. $\qquad\square$

Finally, we prove Theorem 4.2 and Corollary 4.3.

**Theorem 4.2.** *Define $\delta = m/n \leqslant 1$ and let $\rho = m/k$. Consider the attacker $\mathcal{G}^*$ defined by the following generative process: Given a leaked sample $Y \in \mathbb{R}^{m \times d}$, $\mathcal{G}^*$ generates a sample $X \in \mathbb{R}^{n \times d}$ as follows: it first samples $n$ vectors $W_1, \ldots, W_n \overset{iid}{\sim} \mathcal{N}(0, \Sigma)$ and then sets: $X_i = W_i - \bar{W} + \bar{Y}$. Define the authenticator $\mathcal{D}^*$ by:*

$$\mathcal{D}^*(a, x) = I\left[\|\bar{x} - \bar{a}\|_{\Sigma^{-1}}^2 < \frac{d\left(1 + \rho\right)\left(1 + \rho\delta^{-1}\right)}{n(1 - \delta)} \log\left(\frac{\rho + 1}{\rho + \delta}\right)\right] \tag{F.5}$$

*Then $(\mathcal{D}^*, \mathcal{G}^*)$ is a solution of Eq. 2.2 that satisfies Eq. 4.1.*

*Proof.* Directly from Lemma F.8, Corollary F.10, and Theorem F.11 by assigning $\delta = \frac{m}{n}, \rho = \frac{m}{k}$. $\qquad\square$

**Corollary 4.3.** *Define $\delta$ and $\rho$ as in Theorem 4.2. Then the game value for the Gaussian case is:*

$$\frac{1}{2} + \frac{1}{2\Gamma(\frac{d}{2})}\left[\gamma\left(\frac{d}{2}, \frac{d(1 + \rho)}{2(1 - \delta)} \log\frac{1 + \rho}{\delta + \rho}\right) - \gamma\left(\frac{d}{2}, \frac{d(\delta + \rho)}{2(1 - \delta)} \log\frac{1 + \rho}{\delta + \rho}\right)\right] \tag{F.6}$$

*Where $\gamma$ is the lower incomplete Gamma function, and $\Gamma$ is the Gamma function.*

*Proof.* Directly from Theorem F.11 by assigning $\delta = \frac{m}{n}, \rho = \frac{m}{k}$. $\qquad\square$

### F.4 GAME VALUE FOR A MAXIMUM LIKELIHOOD ATTACKER

In this section, we consider the most intuitive attacker strategy, which one could naively see as optimal. However, we show that this intuitive "optimal attacker" is sub-optimal as can be seen in Fig. 1c in the main paper. We consider an attacker that draws the attack sample from a Gaussian distribution with the maximum likelihood estimate of the mean and the known covariance. We denote this attacker by the name ML attacker. We find the best response authenticator to this attacker and the associated game value. Fig. 6 visualizes the difference in theoretical game value between the ML attacker (see Definition F.12) and the optimal attacker (see Definition F.1) for different values of $d$ (the dimension of observations), and demonstrates that the ML attacker is indeed sub-optimal.

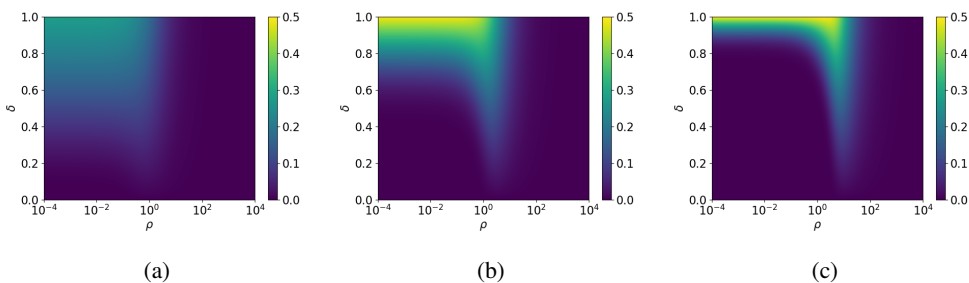

Figure 6: The difference in game value (expected authentication accuracy of the optimal authenticator) between the ML attacker and the optimal attacker for different values of the observations' dimension $d$, as a function of the parameters $\rho = \frac{m}{k}, \delta = \frac{m}{n}$. Namely: $\max_{\mathcal{D}}\{V(\mathcal{D}, \mathcal{G}_{ML})\} - \max_{\mathcal{D}}\{V(\mathcal{D}, \mathcal{G}^*)\}$. (a) Difference in game value for $d = 10$. (b) Difference in game value for $d = 100$. (c) Difference in game value for $d = 1000$.

**Definition F.12.** *Let $\mathcal{G}_{ML}$ denote an attacker defined by the following generative process: Given a leaked sample $Y \in \mathbb{R}^{m \times d}$, $\mathcal{G}_{ML}$ generates an attack sample $X \overset{iid}{\sim} \mathcal{N}(\bar{Y}, \Sigma)$*

**Lemma F.13.** *Let $\theta \in \mathbb{R}^d, \Sigma \in \mathbb{R}^{d \times d}$ represent the mean and covariance of a Gaussian distribution. Let $X \in \mathbb{R}^{n \times d}$ be a random sample generated by the attacker defined in Def. F.12. Then:*

$$X_c \sim \mathcal{N}(\theta_{c,n}, diag(\Sigma, n) + rep((\frac{1}{m})\Sigma, n)) \equiv \mathcal{N}(\theta_{c,n}, \Psi_{ML})$$

*Proof.* Let $W_1, \ldots, W_n \overset{iid}{\sim} \mathcal{N}(0, \Sigma)$, observe that $X_i = \bar{Y} + W_i \quad \forall i \in [n]$, and thus:

$$X_c = \bar{Y}_{c,n} + W_c$$

Where $W_c \sim \mathcal{N}(0 \cdot 1_{dn}, diag(\Sigma, n))$. Using Lemma F.3 we have $\bar{Y} \sim \mathcal{N}(\theta, \frac{1}{m}\Sigma)$ and using Lemma F.4 we have $\bar{Y}_{c,n} \sim \mathcal{N}(\theta_{c,n}, rep(\frac{1}{m}\Sigma, n))$.

Let $Z = \begin{bmatrix} W_c \\ \bar{Y}_{c,n} \end{bmatrix}$ and $B = [I_{nd \times nd} \quad , \quad I_{nd \times nd}]$, then $X_c = W_c + \bar{Y}_{c,n} = BZ$. Note that:

$$Z \sim \mathcal{N}(\begin{bmatrix} 0_{nd} \\ \theta_{c,n} \end{bmatrix}, \begin{bmatrix} diag(\Sigma, n) & 0 \\ 0 & rep(\frac{1}{m}\Sigma, n) \end{bmatrix})$$

and therefore we have:

$$X_c \sim \mathcal{N}(B \begin{bmatrix} 0_{nd} \\ \theta_{c,n} \end{bmatrix}, B \begin{bmatrix} diag(\Sigma, n) & 0 \\ 0 & rep(\frac{1}{m}\Sigma, n) \end{bmatrix} B^T)$$

$$= \mathcal{N}(\theta_{c,n}, diag(\Sigma, n) + rep(\frac{1}{m}\Sigma, n))$$

As required. □

**Lemma F.14.** *Let $\Sigma = CC^T \in \mathbb{R}^{d \times d}$ represent the covariance of a Gaussian distribution, and consider the following covariance matrix: $\Psi_{ML} = diag(\Sigma, n) + rep(\frac{1}{m}\Sigma, n)$. Then:*

$$\Psi_{ML}^{-1} = diag(\Sigma^{-1}, n) - \frac{1}{n+m} rep(\Sigma^{-1}, n)$$

*and the determinant is:*

$$\det(\Psi) = \det(\Sigma)^n (\frac{n+m}{m})^d$$

*Proof.* To find the inverse of $\Psi_{ML}$ we first define:

$$U = \begin{bmatrix} \Sigma \\ \vdots \\ \Sigma \end{bmatrix} \in \mathbb{R}^{nd \times d}, \quad V = \frac{1}{m} \begin{bmatrix} I_d \\ \vdots \\ I_d \end{bmatrix} \in \mathbb{R}^{nd \times d} \tag{F.7}$$

Therefore we have:

$$\Psi_{ML}^{-1} = (diag(\Sigma, n) + rep(\frac{1}{m}\Sigma, n))^{-1}$$

$$= (diag(\Sigma, n) + UV^T)^{-1}$$

$$\overset{(i)}{=} diag(\Sigma, n)^{-1} - diag(\Sigma, n)^{-1} U (I_d + V^T diag(\Sigma, n)^{-1} U)^{-1} V^T diag(\Sigma, n)^{-1}$$

$$\overset{(ii)}{=} diag(\Sigma^{-1}, n) - diag(\Sigma^{-1}, n) U (I_d + V^T diag(\Sigma^{-1}, n) U)^{-1} V^T diag(\Sigma^{-1}, n)$$

$$= diag(\Sigma^{-1}, n) - \begin{bmatrix} I_d \\ \vdots \\ I_d \end{bmatrix} (I_d + \frac{1}{m} [I_d \quad \cdots \quad I_d] \begin{bmatrix} I_d \\ \vdots \\ I_d \end{bmatrix})^{-1} \frac{1}{m} [\Sigma^{-1} \quad \cdots \quad \Sigma^{-1}]$$

$$= diag(\Sigma^{-1}, n) - \begin{bmatrix} I_d \\ \vdots \\ I_d \end{bmatrix} (I_d + \frac{n}{m} I_d)^{-1} \frac{1}{m} [\Sigma^{-1} \quad \cdots \quad \Sigma^{-1}]$$

$$= diag(\Sigma^{-1}, n) - \begin{bmatrix} I_d \\ \vdots \\ I_d \end{bmatrix} ((\frac{n+m}{m}) I_d)^{-1} \frac{1}{m} [\Sigma^{-1} \quad \cdots \quad \Sigma^{-1}]$$

$$= diag(\Sigma^{-1}, n) - \frac{1}{n+m} \begin{bmatrix} I_d \\ \vdots \\ I_d \end{bmatrix} I_d [\Sigma^{-1} \quad \cdots \quad \Sigma^{-1}]$$

$$= diag(\Sigma^{-1}, n) - \frac{1}{n+m} rep(\Sigma^{-1}, n)$$

As required. Where in $(i)$ we used the Woodbury matrix identity, and in $(ii)$ we used the inverse of a diagonal block matrix.

Next, we turn to find the determinant of $\Psi_{ML}$:

$$\det(\Psi) = \det(diag(\Sigma, n) + rep((\frac{1}{m})\Sigma, n))$$

$$= \det(diag(\Sigma, n) + UV^T)$$

$$\overset{(iii)}{=} \det(diag(\Sigma, n)) \det(I_d + V^T diag(\Sigma, n)^{-1} U)$$

$$\overset{(iv)}{=} \det(\Sigma)^n \det(I_d + V^T diag(\Sigma^{-1}, n) U)$$

$$= \det(\Sigma)^n \det(I_d + \frac{1}{m} \begin{bmatrix} I_d & \cdots & I_d \end{bmatrix} diag(\Sigma^{-1}, n) \begin{bmatrix} \Sigma \\ \vdots \\ \Sigma \end{bmatrix})$$

$$= \det(\Sigma)^n \det((\frac{n+m}{m}) I_d)$$

$$= \det(\Sigma)^n (\frac{n+m}{m})^d$$

Where in $(iii)$ we used the matrix determinant lemma, and in $(iv)$ we used the determinant of a diagonal block matrix. $\qquad\square$

**Lemma F.15.** *Consider the attacker $\mathcal{G}_{ML}$, defined in F.12. The best response strategy for the authenticator against this attacker is:*

$$\mathcal{D}_{ML}(a, x) = I\left[\|\bar{x} - \bar{a}\|_{\Sigma^{-1}}^2 < \alpha_{ML}\right]$$

*Where:*

$$\alpha_{ML} = \frac{d(n+k)(nm+nk+mk)}{k^2 n^2} \log(\frac{nm+nk+mk}{m(n+k)})$$

*Proof.* The best response authenticator satisfies:

$$\mathcal{D}^* \in \underset{\mathcal{D} \in \mathbb{D}}{\operatorname{argmax}} V(\mathcal{D}, \mathcal{G}_{ML})$$

$$= \underset{\mathcal{D} \in \mathbb{D}}{\operatorname{argmax}} \frac{1}{2} \mathbb{E}_{\Theta \sim Q} \mathbb{E}_{A \sim f_\Theta^{(k)}} \mathbb{E}_{Y \sim f_\Theta^{(m)}} \left[\mathbb{E}_{X \sim f_\Theta^{(n)}}[\mathcal{D}(A, X)] + \mathbb{E}_{X \sim g_{X|Y}^{ML}(\cdot|Y)}[1 - \mathcal{D}(A, X)]\right]$$

$$= \underset{\mathcal{D} \in \mathbb{D}}{\operatorname{argmax}} \mathbb{E}_{\Theta \sim Q} \mathbb{E}_{A \sim f_\Theta^{(k)}} \left[\mathbb{E}_{X \sim f_\Theta^{(n)}}[\mathcal{D}(A, X)] + \mathbb{E}_{X \sim g_{X|\Theta}^{ML}(\cdot|\Theta)}[1 - \mathcal{D}(A, X)]\right]$$

$$\overset{\text{Lemma F.13}}{=} \underset{\mathcal{D} \in \mathbb{D}}{\operatorname{argmax}} \mathbb{E}_{\Theta \sim Q} \mathbb{E}_{A \sim \mathcal{N}(\Theta_{c,k}, diag(\Sigma, k))}$$

$$\left[\mathbb{E}_{X \sim \mathcal{N}(\Theta_{c,n}, diag(\Sigma, n))}[\mathcal{D}(A, X)] + \mathbb{E}_{X \sim \mathcal{N}(\theta_{c,n}, \Psi_{ML})}[1 - \mathcal{D}(A, X)]\right]$$

$$= \underset{\mathcal{D} \in \mathbb{D}}{\operatorname{argmax}} \mathbb{E}_{\Theta \sim Q} \int_{a \in \mathbb{R}^{kd}} da \int_{x \in \mathbb{R}^{nd}} dx \exp\{-\frac{1}{2}(a - \Theta_{c,k})^T diag(\Sigma, k)^{-1}(a - \Theta_{c,k})\}$$

$$[\sqrt{\frac{1}{|\det(diag(\Sigma, n))|}} \exp\{-\frac{1}{2}(x - \Theta_{c,n})^T diag(\Sigma, n)^{-1}(x - \Theta_{c,n})\} \mathcal{D}(a, x) +$$

$$\sqrt{\frac{1}{|\det(\Psi_{ML})|}} \exp\{-\frac{1}{2}(x - \Theta_{c,n})^T \Psi_{ML}^{-1}(x - \Theta_{c,n})\}[1 - \mathcal{D}(a, x)]]$$

$$\overset{\text{Lemma F.14}}{=} \underset{\mathcal{D} \in \mathbb{D}}{\operatorname{argmax}} \mathbb{E}_{\Theta \sim Q} \int_{a \in \mathbb{R}^{kd}} da \int_{x \in \mathbb{R}^{nd}} dx \exp\{-\frac{1}{2}(a - \Theta_{c,k})^T diag(\Sigma, k)^{-1}(a - \Theta_{c,k})\}[$$

$$\sqrt{(\frac{n+m}{m})^d} \exp\{-\frac{1}{2}(x - \Theta_{c,n})^T diag(\Sigma, n)^{-1}(x - \Theta_{c,n})\} \mathcal{D}(a, x) +$$

$$\exp\{-\frac{1}{2}(x-\Theta_{c,n})^T\Psi_{ML}^{-1}(x-\Theta_{c,n})\}[1-\mathcal{D}(a,x)]]]$$

$\mathcal{D}$ can be chosen independently for each pair $(a,x)\in\mathcal{X}^k\times\mathcal{X}^n$. Therefore, for any $(a,x)\in\mathcal{X}^k\times\mathcal{X}^n$ the decision rule for $\mathcal{D}(a,x)=1$ is:

$$\sqrt{(\frac{n+m}{m})^d}\int_{\theta\in\mathbb{R}^d}d\theta Q(\theta)\exp\{-\frac{1}{2}(x-\theta_{c,n})^T diag(\Sigma,n)^{-1}(x-\theta_{c,n})\}$$

$$\exp\{-\frac{1}{2}(a-\theta_{c,k})^T diag(\Sigma,k)^{-1}(a-\theta_{c,k})\}>$$

$$\int_{\theta\in\mathbb{R}^d}d\theta Q(\theta)\exp\{-\frac{1}{2}(x-\theta_{c,n})^T\Psi_{ML}^{-1}(x-\theta_{c,n})\}\exp\{-\frac{1}{2}(a-\theta_{c,k})^T diag(\Sigma,k)^{-1}(a-\theta_{c,k})\}$$

Observing the LHS integral and using the improper uniform prior assumption, we obtain:

$$\int_{\theta\in\mathbb{R}^d}d\theta Q(\theta)\exp\{-\frac{1}{2}[(x-\theta_{c,n})^T diag(\Sigma,n)^{-1}(x-\theta_{c,n})+$$

$$(a-\theta_{c,k})^T diag(\Sigma,k)^{-1}(a-\theta_{c,k})]\}$$

$$=\int_{\theta\in\mathbb{R}^d}d\theta\exp\{-\frac{1}{2}[\sum_{i=1}^n x_i^T\Sigma^{-1}x_i+\sum_{j=1}^k a_j^T\Sigma^{-1}a_j-2\theta^T\Sigma^{-1}(n\bar{x}+k\bar{a})+(n+k)\theta^T\Sigma^{-1}\theta]\}$$

$$=\exp\{-\frac{1}{2}[\sum_{i=1}^n x_i^T\Sigma^{-1}x_i+\sum_{j=1}^k a_j^T\Sigma^{-1}a_j-(n+k)(\frac{n\bar{x}+k\bar{a}}{n+k})^T\Sigma^{-1}(\frac{n\bar{x}+k\bar{a}}{n+k})]\}$$

$$\int_{\theta\in\mathbb{R}^d}d\theta\exp\{-\frac{n+k}{2}[(\theta-\frac{n\bar{x}+k\bar{a}}{n+k})^T\Sigma^{-1}(\theta-\frac{n\bar{x}+k\bar{a}}{n+k})]\}$$

$$=\exp\{-\frac{1}{2}[\sum_{i=1}^n x_i^T\Sigma^{-1}x_i+\sum_{j=1}^k a_j^T\Sigma^{-1}a_j-(n+k)(\frac{n\bar{x}+k\bar{a}}{n+k})^T\Sigma^{-1}(\frac{n\bar{x}+k\bar{a}}{n+k})]\}$$

$$\sqrt{(\frac{2\pi}{n+k})^d\det(\Sigma)}$$

Observing the RHS and using the improper uniform prior assumption, we obtain:

$$\int_{\theta\in\mathbb{R}^d}d\theta Q(\theta)\exp\{-\frac{1}{2}[(x-\theta_{c,n})^T\Psi_{ML}^{-1}(x-\theta_{c,n})+(a-\theta_{c,k})^T diag(\Sigma^{-1},k)(a-\theta_{c,k})]\}$$

$$=\int_{\theta\in\mathbb{R}^d}d\theta\exp\{-\frac{1}{2}[(x-\theta_{c,n})^T\Psi_{ML}^{-1}(x-\theta_{c,n})+(a-\theta_{c,k})^T diag(\Sigma^{-1},k)(a-\theta_{c,k})]\}$$

$$\overset{\text{F.14}}{=}\int_{\theta\in\mathbb{R}^d}d\theta\exp\{-\frac{1}{2}[(x-\theta_{c,n})^T(diag(\Sigma^{-1},n)-\frac{1}{n+m}rep(\Sigma^{-1},n))(x-\theta_{c,n})+$$

$$(a-\theta_{c,k})^T diag(\Sigma^{-1},k)(a-\theta_{c,k})]\}$$

$$=\int_{\theta\in\mathbb{R}^d}d\theta\exp\{-\frac{1}{2}[\sum_{i=1}^n(x_i-\theta)^T\Sigma^{-1}(x_i-\theta)+\sum_{j=1}^k(a_j-\theta)^T\Sigma^{-1}(a_j-\theta)-$$

$$\frac{n^2}{n+m}(\bar{x}-\theta)^T\Sigma^{-1}(\bar{x}-\theta)]\}$$

$$\overset{(i)}{=}\exp\{-\frac{1}{2}[\sum_{i=1}^n x_i^T\Sigma^{-1}x_i+\sum_{j=1}^k a_j^T\Sigma^{-1}a_j-\frac{n^2}{n+m}\bar{x}^T\Sigma^{-1}\bar{x}-\frac{nm+nk+mk}{n+m}v^T\Sigma^{-1}v]\}$$

$$\int_{\theta\in\mathbb{R}^d}d\theta\exp\{-\frac{1}{2}\frac{nm+nk+mk}{n+m}[v^T\Sigma^{-1}v]\}$$

$$=\exp\{-\frac{1}{2}[\sum_{i=1}^n x_i^T\Sigma^{-1}x_i+\sum_{j=1}^k a_j^T\Sigma^{-1}a_j-\frac{n^2}{n+m}\bar{x}^T\Sigma^{-1}\bar{x}-\frac{nm+nk+mk}{n+m}v^T\Sigma^{-1}v]\}$$

$$\sqrt{(\frac{2\pi(n+m)}{nm+nk+mk})^d\det(\Sigma)}$$

Where in $(i)$ we denoted $v = \frac{nm\bar{x}+k(n+m)\bar{a}}{nm+nk+mk}$. Therefore, the decision rule is

$$\exp\{\frac{1}{2}[(n+k)(\frac{n\bar{x}+k\bar{a}}{n+k})^T\Sigma^{-1}(\frac{n\bar{x}+k\bar{a}}{n+k})]\}\sqrt{(\frac{1}{m(n+k)})^d} >$$

$$\exp\{\frac{1}{2}[\frac{n^2}{n+m}\bar{x}^T\Sigma^{-1}\bar{x} + \frac{nm+nk+mk}{n+m}v^T\Sigma^{-1}v]\}\sqrt{(\frac{1}{nm+nk+mk})^d}$$

$$\Leftrightarrow\sqrt{(\frac{nm+nk+mk}{m(n+k)})^d} >$$

$$\exp\{\frac{1}{2}[\frac{n^2}{n+m}\bar{x}^T\Sigma^{-1}\bar{x} + \frac{nm+nk+mk}{n+m}v^T\Sigma^{-1}v - (n+k)(\frac{n\bar{x}+k\bar{a}}{n+k})^T\Sigma^{-1}(\frac{n\bar{x}+k\bar{a}}{n+k})]\}$$

$$\Leftrightarrow d\log(\frac{nm+nk+mk}{m(n+k)}) >$$

$$\frac{n^2}{n+m}\bar{x}^T\Sigma^{-1}\bar{x} + \frac{nm+nk+mk}{n+m}v^T\Sigma^{-1}v - (n+k)(\frac{n\bar{x}+k\bar{a}}{n+k})^T\Sigma^{-1}(\frac{n\bar{x}+k\bar{a}}{n+k})$$

$$\Leftrightarrow d\log(\frac{nm+nk+mk}{m(n+k)}) > \frac{k^2n^2}{(n+k)(nm+nk+mk)}(\bar{x}-\bar{a})^T\Sigma^{-1}(\bar{x}-\bar{a})$$

$$\Leftrightarrow (\bar{x}-\bar{a})^T\Sigma^{-1}(\bar{x}-\bar{a}) < \frac{d(n+k)(nm+nk+mk)}{k^2n^2}\log(\frac{nm+nk+mk}{m(n+k)})$$

As required. $\square$

**Theorem F.16.** *Fix the attacker to be $\mathcal{G}_{ML}$ as defined in F.12, then the game value is:*

$$\max_{\mathcal{D}} V(\mathcal{D}, \mathcal{G}_{ML}) \overset{Lemma\ F.15}{=} V(\mathcal{D}_{ML}, \mathcal{G}_{ML}) =$$

$$\frac{1}{2} + \frac{1}{2}\frac{1}{\Gamma(\frac{d}{2})}[\gamma(\frac{d}{2}, \frac{d(nm+nk+mk)}{2nk}\log\frac{nm+nk+mk}{m(n+k)}) -$$

$$\gamma(\frac{d}{2}, \frac{dm(n+k)}{2nk}\log\frac{nm+nk+mk}{m(n+k)})] =$$

$$\frac{1}{2} + \frac{1}{2}\frac{1}{\Gamma(\frac{d}{2})}[\gamma(\frac{d}{2}, \frac{d}{2}(1+\rho+\delta)\log\frac{1+\rho+\delta}{\rho+\delta}) - \gamma(\frac{d}{2}, \frac{d}{2}(\rho+\delta)\log\frac{1+\rho+\delta}{\rho+\delta})]$$

*Where $\rho = \frac{m}{k}, \delta = \frac{m}{n}$, and $\gamma$ is the lower incomplete gamma function.*

*Proof.* The game value is given by:

$$V(\mathcal{D}_{ML}, \mathcal{G}_{ML})$$

$$= \mathbb{E}_{\Theta\sim Q}V(\Theta, \mathcal{D}_{ML}, \mathcal{G}_{ML})$$

$$= \frac{1}{2}\mathbb{E}_{\Theta\sim Q}\mathbb{E}_{A\sim f_\Theta^{(k)}}\mathbb{E}_{Y\sim f_\Theta^{(m)}}\left[\mathbb{E}_{X\sim f_\Theta^{(n)}}[\mathcal{D}_{ML}(A, X)] + \mathbb{E}_{X\sim g_{X|Y}^{ML}(\cdot|Y)}[1 - \mathcal{D}_{ML}(A, X)]\right]$$

$$= \frac{1}{2}\mathbb{E}_{\Theta\sim Q}\mathbb{E}_{A\sim f_\Theta^{(k)}}\mathbb{E}_{Y\sim f_\Theta^{(m)}}$$

$$\left[\mathbb{E}_{X\sim f_\Theta^{(n)}}\left[I[\|\bar{X}-\bar{A}\|_{\Sigma^{-1}}^2 < \alpha_{ML}]\right] + \mathbb{E}_{X\sim g_{X|Y}^{ML}(\cdot|Y)}\left[1 - I[\|\bar{X}-\bar{A}\|_{\Sigma^{-1}}^2 < \alpha_{ML}]\right]\right]$$

$$= \frac{1}{2} + \frac{1}{2}\mathbb{E}_{\Theta\sim Q}\mathbb{E}_{A\sim f_\Theta^{(k)}}\mathbb{E}_{X\sim f_\Theta^{(n)}}\left[I[\|\bar{X}-\bar{A}\|_{\Sigma^{-1}}^2 < \alpha_{ML}]\right] -$$

$$\frac{1}{2}\mathbb{E}_{\Theta\sim Q}\mathbb{E}_{A\sim f_\Theta^{(k)}}\mathbb{E}_{X\sim g_{X|\Theta}^{ML}(\cdot|\Theta)}\left[I[\|\bar{X}-\bar{A}\|_{\Sigma^{-1}}^2 < \alpha_{ML}]\right]$$

Observing the first term, we can see that by replacing $\alpha^*$ with $\alpha_{ML}$ in the analog part of the proof for Theorem F.11 we get:

$$\frac{1}{2}\mathbb{E}_{\Theta\sim Q}\mathbb{E}_{A\sim f_\Theta^{(k)}}\mathbb{E}_{X\sim f_\Theta^{(n)}}\left[I[\|\bar{X}-\bar{A}\|_{\Sigma^{-1}}^2 < \alpha_{ML}]\right] = \frac{1}{2}\frac{1}{\Gamma(\frac{d}{2})}\gamma(\frac{d}{2}, \frac{nk\alpha_{ML}}{2(n+k)})$$

Again, similarly to the analog part of the proof for Theorem F.11, observing the second term we have:

$$\frac{1}{2}\mathbb{E}_{\Theta\sim Q}\mathbb{E}_{A\sim f_{\Theta}^{(k)}}\mathbb{E}_{X\sim g_{X|\Theta}^{ML}(\cdot|\Theta)}\left[I[\|\bar{X}-\bar{A}\|_{\Sigma^{-1}}^2<\alpha_{ML}]\right]$$
$$\equiv\frac{1}{2}\mathbb{E}_{\Theta\sim Q}\mathbb{E}_{A\sim f_{\Theta}^{(k)}}\mathbb{E}_{X\sim g_{X|\Theta}^{ML}(\cdot|\Theta)}\left[I[V^TV<\alpha_{ML}]\right]$$
$$=(*)$$

Where:

$$V=C^{-1}(\bar{X}-\bar{A})=C^{-1}[(\bar{X}-\Theta)-(\bar{A}-\Theta)]$$
$$=C^{-1}[I_d,-I_d]\begin{bmatrix}\bar{X}-\Theta\\\bar{A}-\Theta\end{bmatrix}=[C^{-1},-C^{-1}]\begin{bmatrix}\bar{X}-\Theta\\\bar{A}-\Theta\end{bmatrix}$$

Using the definition of $\mathcal{G}_{ML}$ (Definition F.12) we have $\bar{X}\sim\mathcal{N}(\theta,\frac{n+m}{nm}\Sigma)$, using Lemma F.3 we have $\bar{A}\sim\mathcal{N}(\theta,\frac{1}{k}\Sigma)$, and thus:

$$\begin{bmatrix}\bar{X}-\Theta\\\bar{A}-\Theta\end{bmatrix}\sim\mathcal{N}(\begin{bmatrix}\theta\\\theta\end{bmatrix},\begin{bmatrix}\frac{n+m}{nm}\Sigma&0\\0&\frac{1}{k}\Sigma\end{bmatrix})$$

Therefore:

$$V\sim\mathcal{N}(0,[C^{-1},-C^{-1}]\begin{bmatrix}\frac{n+m}{nm}\Sigma&0\\0&\frac{1}{k}\Sigma\end{bmatrix}\begin{bmatrix}C^{-T}\\-C^{-T}\end{bmatrix})=\mathcal{N}(0_d,\frac{nm+nk+mk}{nmk}I_d)$$

We denote $\tilde{V}=\sqrt{\frac{nmk}{nm+nk+mk}}V$ and thus $\tilde{V}_1,\ldots,\tilde{V}_d$ are independent standard normal random variables and $\tilde{V}^T\tilde{V}\sim\chi^2(d)$. Therefore

$$V^TV\sim\Gamma(k=\frac{d}{2},\theta=2\frac{nm+nk+mk}{nmk})$$

And we have:

$$(*)=\frac{1}{2}\frac{1}{\Gamma(\frac{d}{2})}\gamma(\frac{d}{2},\frac{nmk\alpha_{ML}}{2(nm+nk+mk)})$$

Hence, the game value is given by:

$$V(\mathcal{D}^*,\mathcal{G}^*)=\frac{1}{2}+\frac{1}{2}\frac{1}{\Gamma(\frac{d}{2})}[\gamma(\frac{d}{2},\frac{nk\alpha_{ML}}{2(n+k)})-\gamma(\frac{d}{2},\frac{nmk\alpha_{ML}}{2(nm+nk+mk)})]=$$
$$\frac{1}{2}+\frac{1}{2}\frac{1}{\Gamma(\frac{d}{2})}[\gamma(\frac{d}{2},\frac{d(nm+nk+mk)}{2nk}\log\frac{nm+nk+mk}{m(n+k)})-$$
$$\gamma(\frac{d}{2},\frac{dm(n+k)}{2nk}\log\frac{nm+nk+mk}{m(n+k)})]$$

As required. $\qquad\square$

## G  EXPERIMENTS - DATASETS

Below we provide details on the datasets used for the authentication experiments on faces and characters. The VoxCeleb2 (Nagrani et al., 2017; Chung & Zisserman, 2018) dataset contains cropped face videos of 6112 identities. We used the original split of 5994 identities for training and 118 for test. For each identity, we saved every fifth frame, resized each frame to $64\times64$, and augmented it using horizontal flip. The Omniglot dataset (Lake et al., 2015) contains handwritten character images from 50 alphabets. There are 1623 different characters, and 20 examples for each character. We use the splits and augmentations suggested by Vinyals et al. (2016) and used by Snell et al. (2017).

# H    EXPERIMENTS - IMPLEMENTATION DETAILS

In this section, we describe our implementation of the GIM model for the different experiments, in detail. Recall from Sec. 5, that in general, the authenticator is a neural network $\mathcal{D}(a, x)$ that can be expressed as $\mathcal{D}(a, x) = \sigma(T_{\mathcal{D}}(a), T_{\mathcal{D}}(x))$, and the generator is a neural network $\mathcal{G}(y)$ that can be expressed as $\mathcal{G}(y)_i = \varphi(W_i - \bar{W} + T_{\mathcal{G}}(y)) \quad \forall i \in [n]$. In what follows we describe our implementation of these models for each of the experiments.

## H.1    GAUSSIAN SOURCES

**Authenticator Architecture:**  For the statistic function $T_{\mathcal{D}}$, we use a concatenation of the mean and standard deviation of the sample. For the comparison function $\sigma$, we use the element-wise absolute difference between the statistics $T_{\mathcal{D}}(a), T_{\mathcal{D}}(x)$, followed by a linear layer.

**Attacker Architecture:**  For the statistic function $T_{\mathcal{G}}$, we use the sample mean, i.e., $T_{\mathcal{G}}(y) = \bar{y}$. The noise vectors $W_i$ are generated as follows: First, $n$ Gaussian noise vectors $Z_1, \ldots, Z_n \overset{\text{iid}}{\sim} \mathcal{N}(0, I_d)$ are drawn, then each vector $Z_i$ is passed through a linear layer to obtain $W_i$. Finally, the decoder $\varphi$ is the identity function.

**Optimization details:**    The model is trained in an authentication setup as in our theoretical setup, using alternating gradient descent as is common in GAN optimization (Mescheder et al., 2018). Each iteration begins when a source $\theta \in \mathbb{R}^d$ is drawn from the prior distribution $Q = \mathcal{N}(0, 10 I_d)$. Samples $A \in \mathbb{R}^{k \times d}, Y \in \mathbb{R}^{m \times d}, X_\theta \in \mathbb{R}^{n \times d}$ are drawn IID from $f_\theta = \mathcal{N}(\theta, I_d)$, where $X_\theta$ represents a real sample from $\theta$. The attacker, given the leaked sample $Y$, generates a fake sample $X_{\mathcal{G}} = \mathcal{G}(Y) \in \mathbb{R}^{n \times d}$, passes it to $\mathcal{D}$, and suffers the loss $-\log(\text{sigmoid}(\mathcal{D}(A, X_{\mathcal{G}})))$. The authenticator, $\mathcal{D}$, receives as input the source information sample $A$, outputs a prediction for each of the test samples $X_\theta, X_{\mathcal{G}}$, and suffers the binary cross-entropy loss $-0.5 \left(\log\left(\text{sigmoid}(\mathcal{D}(A, X_\theta))\right) + \log\left(\text{sigmoid}(1 - \mathcal{D}(A, X_{\mathcal{G}}))\right)\right)$. Each experiment is trained for $200K$ iterations with a batch size of $4000$ using the Adam optimizer (Kingma & Ba, 2015) with learning rate $10^{-4}$.

## H.2    EXPERIMENTS ON VOXCELEB2 AND OMNIGLOT

To describe the models we begin with some notation. We let $c$ denote the number of image channels, $h$ denote the image size (we only consider square images of size $c \times h \times h$), and $l$ denote the latent dimension of the model.

**Authenticator Architecture:**   As mentioned above, the authenticator is a neural network model that can be expressed as:
$$\mathcal{D}(a, x) = \sigma(T_{\mathcal{D}}(a), T_{\mathcal{D}}(x))$$
The statistic function $T_{\mathcal{D}}$ maps a sample of images to a statistic vector $s \in \mathbb{R}^{6l}$ in the following way: Each image in the sample is mapped using encoders $E_{src}^{\mathcal{D}}, E_{env}^{\mathcal{D}} : [-1, 1]^{c \times h \times h} \to \mathbb{R}^l$ to two latent vectors $v_{src}, v_{env} \in \mathbb{R}^l$, respectively. $v_{src}$ is designed to represent the source $\theta$, and $v_{env}$ is designed to represent the environment (e.g., pose, lighting, expression). To represent the source of the sample, the sample mean of $v_{src}$ is taken. To represent the sample distribution, $v_{env}$ is passed through a non-linear statistic module $\zeta$ which is meant to capture more complex statistical functions of the sample.[5] Finally, $T_{\mathcal{D}}(x)$ is obtained by concatenating $\bar{v}_{src}$ and $\zeta(v_{env})$. E.g., for $x$ we have:

$$T_{\mathcal{D}}(x) = \text{concat} \left( \frac{1}{n} \sum_{i=1}^{n} E_{src}^{\mathcal{D}}(x_i), \zeta \left( E_{env}^{\mathcal{D}}(x) \right) \right)$$

The comparison function $\sigma : \mathbb{R}^{6l} \to \mathbb{R}$ receives two latent vectors $s_a = T_{\mathcal{D}}(a), s_x = T_{\mathcal{D}}(x)$ representing the statistics of the samples $a$ and $x$ respectively. The vectors are concatenated and then passed through a Multi-Layered Perceptron which outputs a scalar reflecting their similarity. Namely:
$$\sigma(s_a, s_x) = \text{MLP} \left( \text{concat} \left( s_a, s_x \right) \right)$$
The full architecture of the authenticator is depicted in Fig. 7.

---

[5]$\zeta$ is implemented as the concatenation of the standard deviation and mean of the sample after passing each example through a Multi-Layered Perceptron.

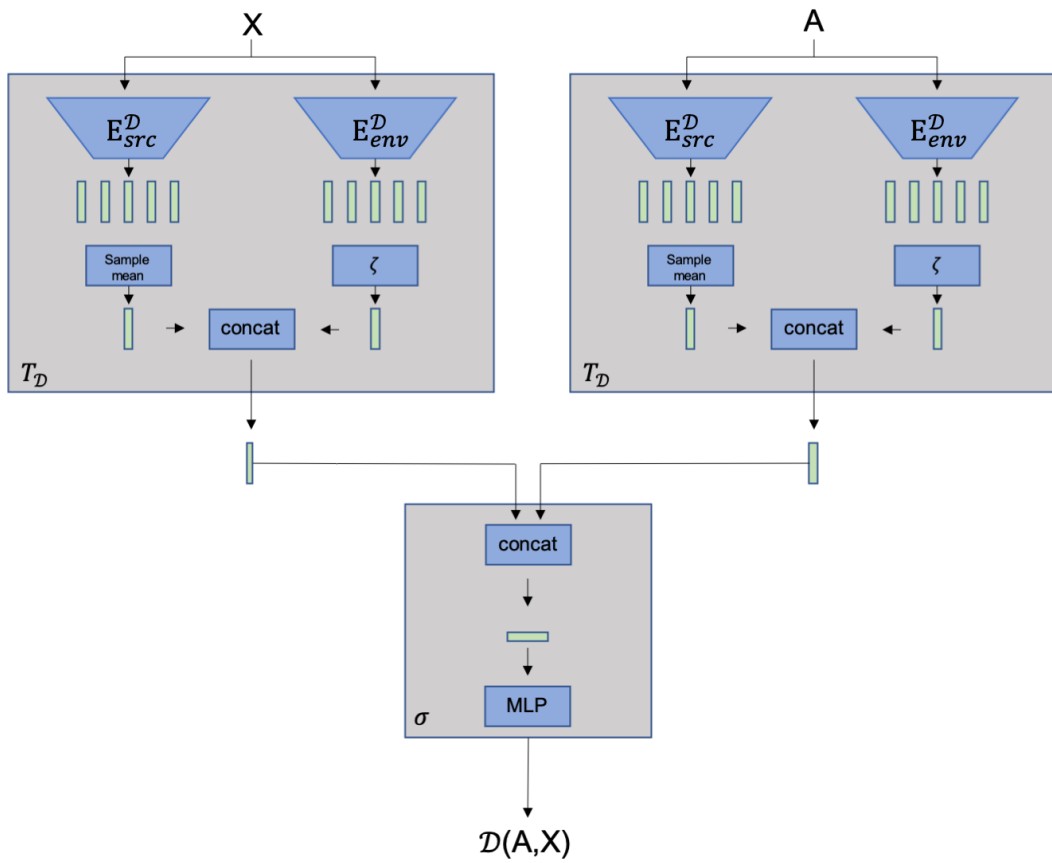

Figure 7: An overview of the implementation of the GIM authenticator architecture for the experiments on the Voxceleb2 and Omniglot datasets.

**Attacker Architecture:** Our implementation of the attacker is inspired by the architecture suggested by Zakharov et al. (2019), which relies on an implicit assumption that an image could be modeled as a mapping of two latent vectors to image space. The first vector represents the source $\theta$ and is the same for any image of $\theta$, the second vector represents the environment (e.g pose, lighting, expression) and is different for each image of the source.

The attacker model consists of the following components: An image encoder $E_{src}^{\mathcal{G}}$ : $[-1,1]^{c \times h \times h} \to \mathbb{R}^l$ that maps an image to a latent vector representing the source $\theta$, an image encoder $E_{env}^{\mathcal{G}} : [-1,1]^{c \times h \times h} \to \mathbb{R}^l$ that maps an image to a latent vector representing the environment, a Multi-layered Perceptron $\text{MLP}_{\mathcal{G}} : \mathbb{R}^l \to \mathbb{R}^l$ that maps Gaussian noise to the environment latent space, an environment decoder $\varphi_{env} : \mathbb{R}^l \to \mathbb{R}^{c \times h \times h}$ that maps a latent vector to an environment image which could represent aspects of the environment such as facial landmarks[6], and finally, a generator $\phi : \mathbb{R}^{2c \times h \times h} \times \mathbb{R}^l \to [-1,1]^{c \times h \times h}$ that maps an environment image concatenated to the real image to a new image. The generator is based on the image to image model used by Zakharov et al. (2019) and Johnson et al. (2016), and uses the source latent vector as input for Adaptive instance normalization (Huang & Belongie, 2017).

The attacker generates a fake sample $X \in [-1,1]^{n \times c \times h \times h}$ based on a leaked sample $Y \in [-1,1]^{m \times c \times h \times h}$ in the following way: Each image $Y_j$ in the leaked sample is mapped using $E_{src}^{\mathcal{G}}$ and $E_{env}^{\mathcal{G}}$ to latent vectors $u_j^{src}, u_j^{env} \in \mathbb{R}^l$. A latent environment vector, $v_i^{env}$, is constructed for each fake image $X_i$ in the following way: First, $n$ Gaussian noise vectors $Z_1, \ldots, Z_n \overset{\text{iid}}{\sim} \mathcal{N}(0, I_l)$

---

[6]In Zakharov et al. (2019), our so-called environment image is indeed a facial landmarks image which is used as input to the model. In our work, we allow the model to learn which environment image is useful.

are drawn, then each vector $Z_i$ is passed through $\mathrm{MLP}_{\mathcal{G}}$ to obtain $W_i$, and finally, $v_i^{env}$ is obtained by matching the mean of the new latent environment vectors to the sample mean $\bar{u}^{env}$. Namely:

$$v_i^{env} = W_i - \bar{W} + \bar{u}^{env} \quad \forall i \in [n]$$

Each fake image $X_i$ is then generated deterministically as follows: $v_i^{env}$ is used as input to the decoder $\varphi_{env}$ which outputs an environment image. This image is concatenated along the channel dimension to a random image from the leaked sample $Y$, and then passed as input to the generator $\phi$, which also receives $u^{src}$ as input to its Adaptive instance norm layers. The output of the generator is the fake image $X_i$ for all $i \in [n]$. The full architecture of the attacker is depicted in Fig. 8.

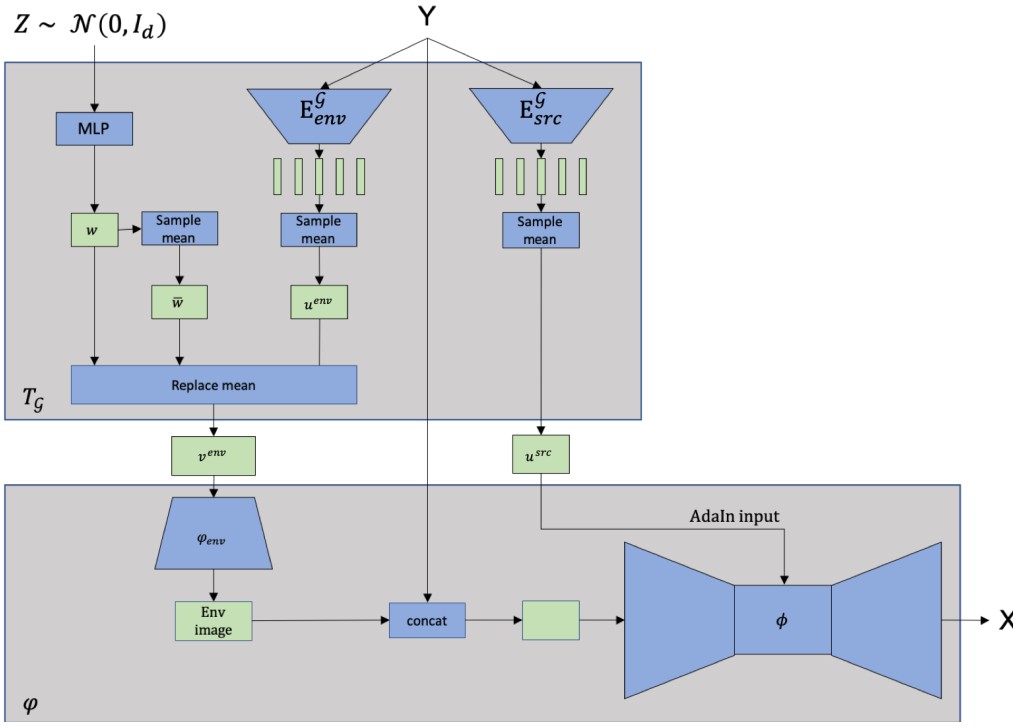

Figure 8: An overview of the implementation of the GIM attacker architecture for the experiments on the Voxceleb2 and Omniglot datasets.

**Optimization details:** The model is trained in an authentication setup as in our theoretical setup, using alternating gradient descent with the regularization parameter as suggested by Mescheder et al. (2018). Each iteration begins when a source $\theta \in \mathbb{R}^d$ is drawn uniformly from the dataset. Samples $A \in [-1,1]^{k \times c \times h \times h}, Y \in [-1,1]^{m \times c \times h \times h}, X_\theta \in [-1,1]^{n \times c \times h \times h}$ are sampled uniformly from the images available to the source $\theta$. The attacker, given the leaked sample $Y$, generates a fake sample $X_{\mathcal{G}} = \mathcal{G}(Y)$, passes it to $\mathcal{D}$, and suffers the loss $-\log(\mathrm{sigmoid}(\mathcal{D}(A, X_{\mathcal{G}})))$. The authenticator, $\mathcal{D}$, receives as input the source information sample $A$, outputs a prediction for each of the test samples $X_\theta, X_{\mathcal{G}}$, and suffers the binary cross-entropy loss $-0.5 \left( \log\left(\mathrm{sigmoid}(\mathcal{D}(A, X_\theta))\right) + \log\left(\mathrm{sigmoid}(1 - \mathcal{D}(A, X_{\mathcal{G}}))\right) \right)$.

The experiments on Omniglot were trained for $520k$ iterations with batch size $128$ using the Adam optimizer (Kingma & Ba, 2015) with learning rate $10^{-6}$ for $\mathcal{D}$, $10^{-5}$ for $\mathcal{G}$, and $10^{-7}$ for $\mathrm{MLP}_{\mathcal{G}}$ (as done by Karras et al. (2018b)). The regularization parameter was set to 0.

The experiments on Voxceleb2 were trained for $250k$ iterations with batch size $64$ using the Adam optimizer (Kingma & Ba, 2015) with learning rate $10^{-4}$ for both $\mathcal{D}$ and $\mathcal{G}$ and $10^{-6}$ for $\mathrm{MLP}_{\mathcal{G}}$. The regularization parameter was set to 10 (as done by Karras et al. (2018b)) since we noticed that it stabilized and sped up the training, and in contrast to Omniglot and the Gaussian experiments did not seem to hurt the results.

