# OpenReview forum: "Optimal Strategies Against Generative Attacks"
_ICLR.cc/2020/Conference — Accept (Talk)_

### Official Review · AnonReviewer2 · 2019-10-24
**Official Blind Review #2**

**Rating:** 8

**Review:**

This paper proposes a new threat model for generative impersonation attacks: The attacker has access to several leaked images of a person; the authenticator knows several registration images per person and decides a person's identify by comparing some newly-sampled images from that person with corresponding registration images. The authors formulate this threat model as a minimax game and analyzed its Nash equilibrium. In the simplified case that observations are multivariate Gaussian, the authors are able to characterize the optimal strategies of the attacker and authenticator explicitly, which gives a nice intuition on how the theoretical optimum changes with respect to data dimension, number of leaked images, etc. Additionally, the authors implemented this attack (named Gan-in-the-middle attack) with an objective similar to GANs, empirically verified the theoretical results, and demonstrated the success of their approach on VoxCeleb2 and

As far as I know, this formulation of generative impersonation attacks is novel. The threat model nicely captures the most important aspects of impersonation attacks and is relatively realistic.

The theoretical analysis is insightful. I especially like that the authors can prove no defense is possible when n <= m, which nicely matches the intuition. The results on Gaussian case not only provide intuition, but also provide motivation for the design of attacker and defender architectures in GIM attacks.

The experiments are well-designed. The model architectures are well-motivated from Theorem 4.2. It is great to see that results of toy experiments match the theoretical analysis in Figure 1(a). The GIM attack on the Voxceleb2 images generates very realistic and reasonable portraits in Figure 2(a). The data augmentation experiment can be naturally fit into the framework of impersonation attacks and the application of their techniques in this direction is very exciting.

I only have two minor suggestions:

1. In Theorem 4.1, the symbol g_{X | Y} was introduced previously, whereas g_{X | A} was never introduced. I have to go to the appendix to understand the definition of g_{X | A}.

2. There is a minor issue in the proof of Lemma D.2 in page 16. The authors seem to miss a 1/2 factor in the second to last row in equation (D.3).





**Experience Assessment:**

I have published one or two papers in this area.

**Review Assessment: Checking Correctness Of Derivations And Theory:**

I assessed the sensibility of the derivations and theory.

**Review Assessment: Checking Correctness Of Experiments:**

I assessed the sensibility of the experiments.

**Review Assessment: Thoroughness In Paper Reading:**

I read the paper at least twice and used my best judgement in assessing the paper.

---

> ### Author Response · Authors · 2019-11-11
> **Thank you very much for your review and great comments.**
>
> Thank you very much for your review and great comments. We address specific comments below:
>
> - Reviewer comment:
> "In Theorem 4.1, the symbol g_{X | Y} was introduced previously, whereas g_{X | A} was never introduced. I have to go to the appendix to understand the definition of g_{X | A}."
>
> - Response:
> The definitions of g_{X | A},f_{X|A} were moved to the appendix due to space considerations. We agree that it should appear in the main text, and have updated the paper accordingly.
>
> - Reviewer comment:
> "There is a minor issue in the proof of Lemma D.2 in page 16. The authors seem to miss a 1/2 factor in the second to last row in equation (D.3)."
>
> - Response:
> Note that the expression in Eq. (D.3) is an argmin, and therefore multiplying or dividing by a constant factor does not change the value of the expression. Therefore, the equality is correct as written. Nevertheless, we agree that it will be clearer to drop the factor 0.5 in the last line - and have updated the paper accordingly.

---

### Official Review · AnonReviewer1 · 2019-10-25
**Official Blind Review #1**

**Rating:** 8

**Review:**

# Summary
The authors investigate an attack-defense problem in which an attacker attempts to pass authentication by generating a faked input, while an authenticator attempts to detect the fraud. They formulate this problem as a zero-sum game and reveal the closed form of the optimal strategies. Furthermore, they reveal a more insightful closed form of the optimal strategies in the Gaussian case. This result clarifies the relationship between the success rate of the attacker and the numbers of the source, registration, and leaked observations. The analysis for the Gaussian case also gives an interesting insight that the optimal attacker’s strategy is to generate fake inputs so that its sufficient statistics are matched to that of the leaked observations. Based on this insight, the authors propose a new learning algorithm for the authenticator and demonstrate by some empirical evaluations that the proposed algorithm is robust against the faked input.

# Detailed comments
This is an interesting and well-written paper. I recommend acceptance of this paper.
The authors investigate an attack of generating a faked input for passing the authenticator under which the attacker can only observe partial information about the source input. This is an interesting point of view and allows us to analyze a more practical situation. Furthermore, based on the theoretical analyses, they reveal an interesting insight of the optimal attacker’s strategy that the optimal strategy generates a faked input so that its sufficient statistics are matched to that of the leaked observations. This insight introduces the new robust learning algorithm which outperforms the existing robust learning algorithm demonstrated as in the empirical evaluations.
Some minor refinements would improve the paper:
- \bar{x} and \bar{a} in Theorem 4.2 should be clearly defined.
- It seems to me that the authors use the term “ML attacker” to denote some different attacking algorithms.

**Experience Assessment:**

I have read many papers in this area.

**Review Assessment: Checking Correctness Of Derivations And Theory:**

I did not assess the derivations or theory.

**Review Assessment: Checking Correctness Of Experiments:**

I assessed the sensibility of the experiments.

**Review Assessment: Thoroughness In Paper Reading:**

I made a quick assessment of this paper.

---

> ### Author Response · Authors · 2019-11-11
> **Thank you very much for your review and great comments.**
>
> Thank you very much for your review and great comments. We address specific comments below:
>
> - Reviewer comment:
> "\bar{x} and \bar{a} in Theorem 4.2 should be clearly defined"
>
> - Response:
> The definition for \bar{x},\bar{a} was indeed missing from the main paper. We have updated the paper to include the definition in section 4.3
>
> - Reviewer comment:
> "It seems to me that the authors use the term “ML attacker” to denote some different attacking algorithms."
>
> - Response:
> As described in Section 4.3, we use the term “ML Attacker” to denote an attacker that uses the intuitive strategy of estimating the mean of the source using maximum likelihood (Hence the name ‘ML’) and generating the attack sample by drawing n iid points from a Gaussian distribution with the estimated mean and known variance. In our setup this strategy turns out to be sub-optimal, as we show theoretically in section F.4 and empirically in Figure 1C.

---

### Official Review · AnonReviewer4 · 2019-11-01
**Official Blind Review #4**

**Rating:** 8

**Review:**

"Optimal Strategies Against Generative Attacks" describes just what the title implies - various dimensions of the problem of defending against a generative adversary, with theoretical discussion under limited settings, as well as practical experiments extending on the intuitions gained using the theoretical exploration under limited conditions.

Particularly, one of the key stated goals of the paper is to "construct a theorectical framework for studying the security risk arising from generative models, and explore its practical implications". Given this goal, the paper performs admirably.

The appendix is extensive, and gives a lot more insight into the core paper itself.

I am unclear on how the data augmentation experiment fits into the overall picture - perhaps a more detailed explanation of how and why this would be used to form an "attack" would help. The other experiments are sensible, and demonstrate reasonable and expected results.

This is a solid paper, and most of my critiques are "out of scope" and revolve around experiments that would be nice to see. Though GAN-for-text is not simple, showing this type of setup for text would be interesting for a number of reasons, same for audio.

Continual passes through the text, with a focus on clarity could also be helpful - the topic is dense, and the text does a good job describing what is happening, but it is always possible to further distill these complex topics, and relegate some useful-but-not-critical pieces to the appendix.

These are minor quibbles, and overall this paper was an interesting and useful read, on a relatively underexplored topic. It shows theorectical results, and practical experimental demonstrations of the theories proposed. Given the stated goals of the paper, it performs admirably.

**Experience Assessment:**

I have published one or two papers in this area.

**Review Assessment: Checking Correctness Of Derivations And Theory:**

I assessed the sensibility of the derivations and theory.

**Review Assessment: Checking Correctness Of Experiments:**

I assessed the sensibility of the experiments.

**Review Assessment: Thoroughness In Paper Reading:**

I read the paper at least twice and used my best judgement in assessing the paper.

---

> ### Author Response · Authors · 2019-11-11
> **Thank you very much for your review and great comments.**
>
> Thank you very much for your review and positive comments. We address specific comments below:
>
> - Reviewer comment:
> "I am unclear on how the data augmentation experiment fits into the overall picture - perhaps a more detailed explanation of how and why this would be used to form an "attack" would help."
>
> - Response:
> The GIM architecture receives as input a set of m iid examples (e.g., images) drawn from a source distribution, and outputs a set of n > m examples that “are similar” to those drawn iid from the same source distribution.
> Thus, GIM can be also used for data augmentation in the context of few-shot learning, where one receives a (usually small) set of examples of a class and wants to generate more instances of this class for training a classifier. We included these experiments in order to show that GIM performs well in this setting, and can thus be used in applications beyond authentication.
>
> - Reviewer comment:
> "This is a solid paper, and most of my critiques are "out of scope" and revolve around experiments that would be nice to see. Though GAN-for-text is not simple, showing this type of setup for text would be interesting for a number of reasons, same for audio."
>
> - Response:
> The idea to extend the current work to signals such as text, audio and video is indeed very interesting, and we plan to do follow up work in these directions: both theoretical (e.g. extend the iid setting to sources that generate stochastic sequences with assumptions such as Markov chains or Hidden Markov models), as well as applied (e.g., training GIM for text, audio and video).
>
> - Reviewer comment:
> "Continual passes through the text, with a focus on clarity could also be helpful - the topic is dense, and the text does a good job describing what is happening, but it is always possible to further distill these complex topics, and relegate some useful-but-not-critical pieces to the appendix."
>
> - Response:
> We will further edit and restructure the text as you suggested before the camera ready version.

---

### Official Review · AnonReviewer3 · 2019-11-03
**Official Blind Review #3**

**Rating:** 8

**Review:**

This paper addresses the issue of malicious use of generative models to fool authentication/anomaly detection systems that rely on sensor data. The authors formulate the scenario as a maxmin game between an authenticator and an attacker, with limitations on the number of samples available to the authenticator to fix a decision rule, the number of samples required at test time for the authenticator to take a decision and the number of leaked samples the attacker has access to. The authors prove that the game admits a Nash equilibrium and derive a closed form solution for the case of multivariate Gaussian data. Finally, the authors propose an algorithm called "GAN In the Middle" and perform experiments to show consistency with the theoretical results, better authentication performance than state of the art methods and usability for data augmentation.

This pager should be accepted. Overall, it addresses crucial problems with the recent advances of generative models and provides significant theoretical results. The experiments would benefit from some clarification.

For the experiments, the following should be addressed:
* Confidence intervals for all results (specifically in Figure 1a and in Table 1)
* Figure 1a: it would be interesting to see a similar analysis also for other values of m and k.
* Table 1: This result would also be more supporting if experiments were performed for varying values of m, n and k. The description of the RS attack could be made more precise: does it mean that the attacker samples images at random? Intuitively, it feels confusing that the GIM authenticator would perform worse on this setting.
* The experiments on handwritten data would be more similar to what I imagine being a real-world authentication scenario  if performed on a task where a class is a single person writing multiple characters, as opposed to characters being classes.


Minor comments:
* Page 6, second to last row: there is a"the" repeated twice.

**Experience Assessment:**

I have read many papers in this area.

**Review Assessment: Checking Correctness Of Derivations And Theory:**

I did not assess the derivations or theory.

**Review Assessment: Checking Correctness Of Experiments:**

I assessed the sensibility of the experiments.

**Review Assessment: Thoroughness In Paper Reading:**

I read the paper at least twice and used my best judgement in assessing the paper.

---

> ### Author Response · Authors · 2019-11-11
> **Thank you very much for your review and great comments.**
>
> Thank you very much for your review and great comments. We address specific comments below:
>
> - Reviewer comment:
> "Confidence intervals for all results (specifically in Figure 1a and in Table 1)"
>
> - Response:
> We will add confidence intervals for the experiments in the camera ready version (these require tens of repetitions per experiment, and are thus time consuming). Our experience with running several restarts and evaluating different checkpoints shows that results are stable, and the trends shown are robust.
>
> - Reviewer comment:
> "Figure 1a: it would be interesting to see a similar analysis also for other values of m and k."
>
> - Response:
> We agree that such experiments are interesting, and were left out due to space considerations. We can definitely add some more Gaussian experiments over different m and k values to the appendix in the camera ready version.
>
> - Reviewer comment:
> "Table 1: This result would also be more supporting if experiments were performed for varying values of m, n and k."
>
> - Response:
> We did perform other experiments (not reported in order to not make the paper too dense) for other values, and observed that the larger the ratio n/m is, the better our authenticator performs (in agreement with our theory). We chose to report results on m=1,n=5,k=5 for simplicity and in order to show that even for very practical n values - our model performs well.
> We will be happy to add to more experimental results for different m,n,k values in the camera ready version.
>
> - Reviewer comment:
> "The description of the RS attack could be made more precise: does it mean that the attacker samples images at random?"
>
> - Response:
> The RS attacker is defined as follows: It first draws a source from a uniform distribution over the sources in the dataset. Then it draws n images of that source from a uniform distribution over the images available to that source.
> One can think of this setting as a scenario where an attacker tries to impersonate a source (e.g. person) without having leaked information on the source’s identity. Thus the attacker chooses an identity at random and uses n real observations of that person.
> We have updated the paper to make the definition more precise.
>
> - Reviewer comment:
> "Intuitively, it feels confusing that the GIM authenticator would perform worse on this setting (RS)."
>
> - Response:
> Note that the GIM authenticator actually achieves better accuracy against the RS attacker than against the GIM attacker on both Omniglot and Voxceleb2. This is exactly what we expect, Since the GIM attacker is expected to be the worst case attack - and thus result in the worst authentication accuracy.
>
> - Reviewer comment:
> "The experiments on handwritten data would be more similar to what I imagine being a real-world authentication scenario  if performed on a task where a class is a single person writing multiple characters, as opposed to characters being classes."
>
> - Response:
> We agree that this is a more realistic authentication setting and thus an interesting experiment to consider. We are not aware of a good existing dataset for this setup, and thus chose Omniglot since it is a commonly used dataset for generative models and few shot classification. However, in followup work we intend to further explore available datasets (e.g., in voice and hand-writing) that capture an authentication setting
>
> - Reviewer comment:
> "Page 6, second to last row: there is a"the" repeated twice."
>
> - Response:
> Thanks for noticing. We updated the paper to fix this typo.

---

### Decision · Program_Chairs · 2019-12-19

**Decision:**

Accept (Talk)

**Comment:**

This paper concerns the problem of defending against generative "attacks": that is, falsification of data for malicious purposes through the use of synthesized data based on "leaked" samples of real data. The paper casts the problem formally and assesses the problem of authentication in terms of the sample complexity at test time and the sample budget of the attacker. The authors prove a Nash equillibrium exists, derive a closed form for the special case of multivariate Gaussian data, and propose an algorithm called GAN in the Middle leveraging the developed principles, showing an implementation to perform better than authentication baselines and suggesting other applications.

Reviewers were overall very positive, in agreement that the problem addressed is important and the contribution made is significant. Most criticisms were superficial. This is a dense piece of work, and presentation could still be improved. However this is clearly a significant piece of work addressing a problem of increasing importance, and is worthy of acceptance.